# Answer, Assemble, Ace: Understanding How LMs Answer Multiple Choice Questions

**Sarah Wiegreffe**[♡♣]    **Oyvind Tafjord**[♡]    **Yonatan Belinkov**[♢]
**Hannaneh Hajishirzi**[♡♣]    **Ashish Sabharwal**[♡]

[♡]Allen Institute for AI, [♣]University of Washington, [♢]Technion
wiegreffesarah@gmail.com

## Abstract

Multiple-choice question answering (MCQA) is a key competence of performant transformer language models that is tested by mainstream benchmarks. However, recent evidence shows that models can have quite a range of performance, particularly when the task format is diversified slightly (such as by shuffling answer choice order). In this work we ask: *how do successful models perform formatted MCQA?* We employ vocabulary projection and activation patching methods to localize key hidden states that encode relevant information for predicting the correct answer. We find that the prediction of a specific answer symbol is causally attributed to a few middle layers, and specifically their multi-head self-attention mechanisms. We show that subsequent layers increase the probability of the predicted answer symbol in vocabulary space, and that this probability increase is associated with a sparse set of attention heads with unique roles. We additionally uncover differences in how different models adjust to alternative symbols. Finally, we demonstrate that a synthetic task can disentangle sources of model error to pinpoint when a model has learned formatted MCQA, and show that logit differences between answer choice tokens continue to grow over the course of training.[1]

## 1 Introduction

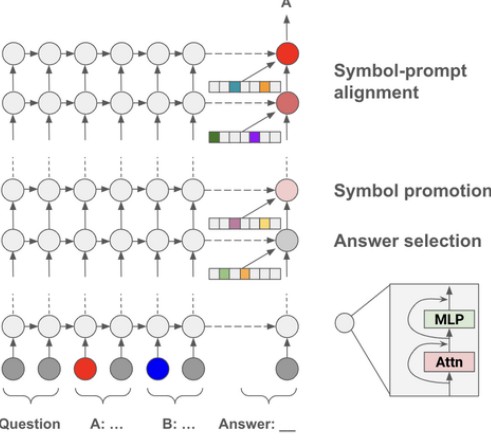

Figure 1: We investigate the ability of transformer LMs to answer formatted multiple-choice questions, which involves producing an answer choice symbol (here, A or B). We discover 1-3 middle layers at the last token position, and particularly their multi-head self-attention functions, responsible for answer selection. Later layers assign increasing probability to the symbol of interest in the model's vocabulary space, for which a sparse set of attention heads are responsible. Finally, when the prompt contains unusual answer choice symbols such as Q/Z/R/X, some models initially assign high values to common answer symbols like A/B/C/D before aligning to the symbols in the prompt at a late layer.

Multiple-choice question answering (MCQA) is a mainstay of language model (LM) evaluation (Liang et al., 2022; Gao et al., 2023; Beeching et al., 2023), not in the least because it avoids the challenges of open-ended text evaluation. A prominent example is the MMLU benchmark (Hendrycks et al., 2021), which is considered a meaningful signal for new model releases (Lambert, 2024a;b).

---

[1]Code is available at https://github.com/allenai/understanding_mcqa.

While early LMs were evaluated on MCQA questions by treating the task as next-token-prediction and scoring each answer choice appended to the input independently, many benchmarks and datasets now evaluate models on their ability to produce the correct answer choice symbol in a single inference run when given the answer choices in the prompt (*formatted MCQA*; Figure 1 and §3.1). Indeed, it has been shown that capable models, and particularly those that have been instruction-tuned, perform well on formatted MCQA tasks (Liang et al., 2022; Robinson & Wingate, 2023; Wiegreffe et al., 2023). At the same time, despite their strong end-task performance, some models are not robust to seemingly inconsequential format changes, such as shuffling the position of the correct answer choice, changing letters that answers are mapped to, or using different symbols (Pezeshkpour & Hruschka, 2024; Alzahrani et al., 2024; Khatun & Brown, 2024).

Understanding how LMs form their predictions, particularly on real-world benchmarks and task formats such as MCQA, is important for reliability reasons. Motivated by this, we ask: *How do successful models perform formatted MCQA?*

In this paper, we interpret internal components to uncover how models promote the prediction of specific answer choice symbols. We perform vocabulary projection and activation patching on three model families—Llama 3.1 (Dubey et al., 2024), Olmo 0724 (Groeneveld et al., 2024) and Qwen 2.5 (Yang et al., 2024)—and three datasets—MMLU (Hendrycks et al., 2021), HellaSwag (Zellers et al., 2019), and a copying task we create. Our findings, summarized in Figure 1, are:

1. When models are correct, they both encode information needed to predict the correct answer symbol and promote answer symbols in the vocabulary space in a very similar fashion across tasks, even when their overall task performance varies.
2. Answer symbol production is driven by a sparse portion of the network, namely, attention heads.
3. The process for correctly answering more unnatural prompt formats is in some cases more complex: Olmo 7B Instruct and Qwen 2.5 1.5B models sometimes only begin producing the correct symbols for these prompts in later layers. We discover that the models' hidden states initially assign high logit values to *expected* answer symbols (here, A/B/C/D) before switching to the symbols given in the prompt (here, random letters like Q/Z/R/X or O/E/B/P).
4. A simple synthetic task can disentangle formatted MCQA performance from dataset-specific performance, and allows us to narrow down a point during training at which Olmo 7B learns formatted MCQA.

## 2 RELATED WORK

Behavioral analyses of LM abilities to answer formatted MCQA questions prompt models in a black-box manner by constructing different input prompts and observing how they affect models' predictions (Wiegreffe et al., 2023; Pezeshkpour & Hruschka, 2024; Sun et al., 2024; Zheng et al., 2024; Alzahrani et al., 2024; Khatun & Brown, 2024; Balepur et al., 2024; Wang et al., 2024). These methods and findings are complementary to ours.

Efforts to interpret transformer LMs mechanistically[2] for MCQA are limited. Most similar to ours is Lieberum et al. (2023), who study what attention heads at the final token position are doing, and isolate a subset they coin "correct letter heads". They demonstrate how these heads attend to answer symbol representations at earlier token positions and promote the correct symbol based on its position in the symbol order, though they show this behavior does not hold entirely for answer choices other than A/B/C/D. Their experiments are limited to one task (MMLU) and one closed-source model (Chinchilla-70B). Apart from broadening the scope of analysis substantially, we also analyze model behavior over the course of training and when it is poor, and create a synthetic task to disentangle task-relevant knowledge from the ability to answer formatted MCQA questions. We additionally put forward a hypothesis for how models adapt when other answer choice symbols are used.

Li & Gao (2024) use vocabulary projection and study the norms of weighted value vectors in multi-layer perceptrons of GPT-2 models in order to localize the production of A, which GPT-2 models are biased towards producing when prompted 0-shot. They find that updating a single value vector can substantially reduce this label bias. Unlike that work, we investigate how models do symbol binding

---

[2]In the narrow technical sense of the term (Saphra & Wiegreffe, 2024).

when they are performant at formatted MCQA; label bias is *one* reason why a model may not be correctly performing symbol binding.

Outside of MCQA, many works investigate how models internally build up representations that lead to their predictions, such as on factual recall (Geva et al., 2021; Dai et al., 2022; Geva et al., 2022b; Meng et al., 2022; Geva et al., 2023; Yu et al., 2023; Merullo et al., 2024a) or basic linguistic (Vig et al., 2020; Wang et al., 2023; Merullo et al., 2024b; Prakash et al., 2024; Todd et al., 2024) or arithmetic (Stolfo et al., 2023; Hanna et al., 2023; Wu et al., 2023) tasks. MCQA differs because it is both directly represented in real benchmarks and involves multiple reasoning steps.

## 3 FORMATTED MCQA

### 3.1 TASK NOTATION

Formatted MCQA is a prompt format in which possible answer choices are presented to the model as an enumerated list, each associated with an **answer choice symbol**. All of our dataset instances are either 0-shot or 3-shot with 4 answer choices, formatted as follows:

```
For each of the following phrases, select the
best completion.

<optional in-context examples of the same format>

Phrase: Corn is yellow. What color is corn?
Choices:
A. yellow
B. grey
C. blue
D. pink
The correct answer is:
```

where $y^* =$ A, B, C, or D (A in this example).[3] Models parameterized by $\theta$ are evaluated on their ability to assign higher probability to the symbol associated with the correct answer choice (here, A) as opposed to the alternative symbols (here, B, C, and D). We compute $\hat{y} = \text{argmax}_{\text{A,B,C,D}} \left[ p_\theta(\text{A}|x), p_\theta(\text{B}|x), p_\theta(\text{C}|x), p_\theta(\text{D}|x) \right]$. To perform well at this task, models must not only predict the correct answer phrase (yellow in the above example), but then map it to its corresponding symbol (A). Robinson & Wingate (2023) refers to this capability as **symbol binding**.

### 3.2 DATASETS

We experiment on two challenging real-world multiple-choice datasets from LLM benchmarks: **HellaSwag** (Zellers et al., 2019) and **MMLU** (Hendrycks et al., 2021). Details are given in Appendix A.1. We additionally use a prototypical colors dataset (Norlund et al., 2021) to create a synthetic 4-way task disentangling dataset-specific knowledge from the ability to perform symbol binding: **Copying Colors from Context ('Colors')**. The dataset consists of instances such as $x =$ Corn is, $y =$ yellow. We include $y$ in the context, so the model's only task is to produce the symbol associated with that answer choice given three distractors randomly selected from the ten colors present in the dataset (i.e., perform symbol binding). We use 3 instances as in-context examples and the remaining 105 as our test set. An example formatted instance is given above.

We validate that all models achieve 100% 3-shot accuracy on a generative version of the Colors task: we provide instances in the format `` Corn is yellow. What color is corn?'' and take the first greedy-decoded token as the prediction (i.e., we expect the model to produce ``yellow'').

### 3.3 MODELS

**Olmo family.** We experiment on the base (Olmo 0724 7B) and instruction-tuned (Olmo 0724 7B Instruct) versions of the most recent (0724) release of the Olmo model (Groeneveld et al., 2024). These have 32 layers with 32 attention heads per layer. We additionally benchmark performance on the smaller Olmo 0724 1B and supervised finetuned Olmo 0724 7B SFT models.

**Llama family.** We experiment with the smallest Llama 3.1 models: Llama 3.1 8B base model and Llama 3.1 8B Instruct (Dubey et al., 2024). These have 32 layers with 32 attention heads per layer.

---

[3]The actual answer tokens include preceding spaces in some of the tokenizers of the models we study.

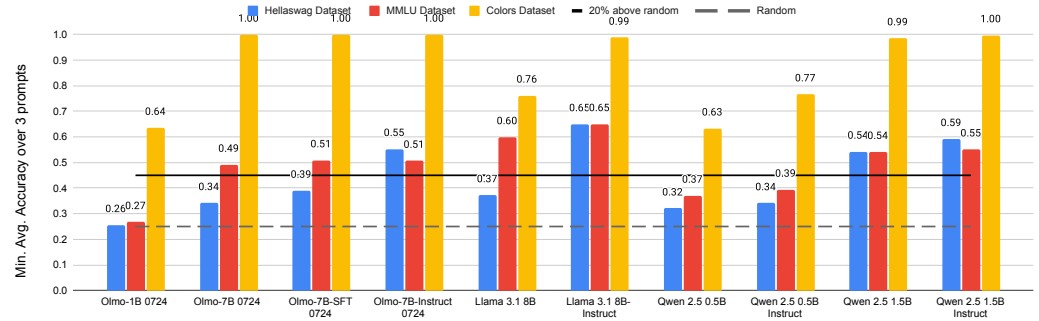

Figure 2: Results by model on Colors, Hellaswag and MMLU. Plotted is the minimum accuracy across `A/B/C/D`, `Q/Z/R/X`, and `1/2/3/4` prompts, where the accuracy for each prompt is taken as the average over all four correct answer positions. 0-shot results for select models in Fig. 11.

**Qwen family.** To study smaller performant models, we include the base and instruct versions of the 0.5B and 1.5B Qwen 2.5 models (Yang et al., 2024). The 1.5B model has 28 layers with 12 attention heads per layer.

### 3.4 IDENTIFYING CONSISTENT MCQA MODELS

We next isolate models capable of performing formatted MCQA by testing whether models answer formatted MCQA prompts *consistently*, i.e., they are robust to the position and symbol of the correct answer choice. For each dataset instance, we construct four versions where we vary the location of the correct answer string, and thus $y^*$. We denote these as **A**/B/C/D, A/**B**/C/D, A/B/**C**/D, and A/B/C/**D**, where bold indicates the location of the correct answer. We average over these four prompts when reporting accuracy, and refer to this collective set of instances of size $4 * |\text{dataset}|$ as `A/B/C/D` prompts.[4] Lastly, to understand what behaviors we localize are specific to the letters used, we additionally include prompts `Q/Z/R/X` and `1/2/3/4`.[5] We evaluate each model per dataset instance on this set of 12 total prompts.

Results are in Fig. 2, where the minimum performance across `A/B/C/D`, `Q/Z/R/X`, and `1/2/3/4` is plotted. Models exhibit differing performance on the 3 datasets, indicating the datasets capture a meaningful range of difficulty. We additionally observe that many models achieve perfect or near-perfect performance on Colors while being far from it on the other two datasets, indicating that error can be attributed to dataset difficulty as opposed to symbol binding. Finally, most (but not all) models are capable of performing symbol binding on the Colors dataset with all 3 sets of answer choice symbols, despite the `Q/Z/R/X` prompt being far less likely to occur in the training data.[6]

We select the best-performing model from each model family for subsequent analysis: Olmo 0724 7B Instruct, Llama 3.1 8B Instruct, and Qwen 2.5 1.5B. They perform all 3 tasks sufficiently above random accuracy (>20% above random, when averaged across prompts and answer positions).

## 4 LOCALIZING ANSWER SYMBOL PRODUCTION

### 4.1 ACTIVATION PATCHING

Activation patching, sometimes referred to as causal tracing (Meng et al., 2022), causal mediation analysis (Vig et al., 2020), or interchange intervention (Geiger et al., 2020), is a method for performing mediation on neural network hidden states to localize which network components have a causal effect on a model's prediction. The method involves performing the following steps:

---

[4] Always predicting a single letter results in 25% accuracy when averaged across correct answer positions. This ensures that models with a strong bias towards a particular symbol are penalized.

[5] When this prompt format is used, we format the in-context examples with the same symbols.

[6] For models that do not perform well on the Colors task, this is due to the MCQA format, since all models can copy and produce the color string from the context with 100% accuracy (§3.2).

1. Run inference on a dataset instance that the model predicts correctly, say, one formatted as **A**/B/C/D ($x_A$). This will produce scores $sc(\text{A}|x_A)$ (higher) and $sc(\text{B}|x_A)$, $sc(\text{C}|x_A)$, $sc(\text{D}|x_A)$ (lower).
2. Run inference on the same dataset instance, but in a different format, such as A/**B**/C/D, that the model still predicts correctly ($x_B$). This will produce a second set of scores $sc(\text{B}|x_B)$ (higher) and $sc(\text{A}|x_B)$, $sc(\text{C}|x_B)$, $sc(\text{D}|x_B)$ (lower). We store hidden state activations of interest.
3. While running inference on $x_A$, replace the output hidden state at the final token position $T$ and layer $\ell$, $\mathbf{h}_{\ell,T}^{(A)}$, with the hidden state from the same layer and token position, but from the inference run of $x_B$, $\mathbf{h}_{\ell,T}^{(B)}$. We measure and plot $sc(\text{A}|x_A, \mathbf{h}_{\ell,T}^{(B)})$, $sc(\text{B}|x_A, \mathbf{h}_{\ell,T}^{(B)})$, $sc(\text{C}|x_A, \mathbf{h}_{\ell,T}^{(B)})$, and $sc(\text{D}|x_A, \mathbf{h}_{\ell,T}^{(B)})$.
4. Repeat step 3 for each layer ($\ell \in [1, L]$).

We use the notation $x_B \to x_A$ to indicate patching a hidden state from $x_B$ into $x_A$. Meng et al. (2022) proposed performing activation patching not only on layerwise outputs, but also on the outputs of multi-layer perceptron (MLP) and multi-head self-attention (MHSA) functions; Todd et al. (2024) extend the latter by patching the outputs of each weighted attention head.[7] We do the same. Note two preliminary conditions for activation patching: the model predicts the correct answer for both $x_A$ and $x_B$, and $x_A$ and $x_B$ do not have the same answer. The score $sc$ can be either a logit or probit value; we discuss this in §4.3.

## 4.2 VOCABULARY PROJECTION

Residual connections have been shown to allow for the iterative refinement of features in neural networks (Jastrzebski et al., 2018; Simoulin & Crabbé, 2021). In addition to understanding which hidden states have the largest causal effect on predictions, it is useful to understand how predictions form in the vocabulary space defined by the dot product between the hidden state output by the LM, $\mathbf{h}_{L,T}$, and the unembedding matrix, $W_U$. Recall that for an $L$-layer autoregressive LM, model predictions are derived from logit values assigned to each item in the vocabulary $\mathcal{V}$, $\mathbf{sc} \in \mathbb{R}^{|\mathcal{V}|}$, given by: $\mathbf{sc} = W_U \cdot \text{LN}(\mathbf{h}_{L,T}))$ where LN is layer normalization, and $\mathbf{h}_{L,T}$ is the hidden state output by the final layer at the last token position. The scores are then further normalized into probits that sum to 1 over the vocabulary: $\mathbf{p} = \text{Softmax}(\mathbf{sc})$.

Prior work (nostalgebraist, 2020; Geva et al., 2021; 2022a;b; Dar et al., 2023; Katz & Belinkov, 2023; Yu et al., 2023; Merullo et al., 2024a, *i.a.*) has proposed projecting *any* hidden state in the Transformer block of dimensionality $d$ to the vocabulary space, in order to inspect when and how models build up to $\mathbf{sc}$ and/or $\mathbf{p}$ via their internal representations. This technique, "vocabulary projection", also sometimes called "logit lens" when logits are used, is equivalent to early exiting on the Transformer block at inference time (Dehghani et al., 2019; Elbayad et al., 2020; Schuster et al., 2021; 2022, *i.a.*). We plot the values assigned to the answer symbol tokens, as well as the maximum values in $\mathbf{sc}$ and $\mathbf{p}$ assigned to any other token, at model states from each layer at the final token position $T$. The "max other token" line delineates when scores assigned to the tokens of interest reach the top of the vocabulary ranking. The "logit difference" line plots the gap between the predicted token and the next-largest of the answer choices (akin to the scoring rule described in §3.1).

## 4.3 LOGITS VS. PROBITS

Logits are a useful choice of metric for activation patching and vocabulary projection, in part because they are capable of detecting model components which demote particular answer choices by assigning lower-than-average logit values (Zhang & Nanda, 2024); these can become indistinguishable from average logit values (i.e., both on or near 0) after Softmax normalization. They also measure the direct additive contribution that each model component makes to the residual stream, and thus have a loosely causal interpretation, assuming no layer normalization on model outputs (Lieberum et al., 2023; Zhu et al., 2024). On the other hand, it is valuable to measure scores assigned to the tokens of interest (such as answer symbol tokens) *with respect to scores assigned to other tokens in the vocabulary* and on a normalized 0-1 scale, as probits do. This 1) gives us insight into the rank

---

[7]MHSA output is a weighted sum of individual attention heads. See Appendix A.2 for background on the Transformer architecture.

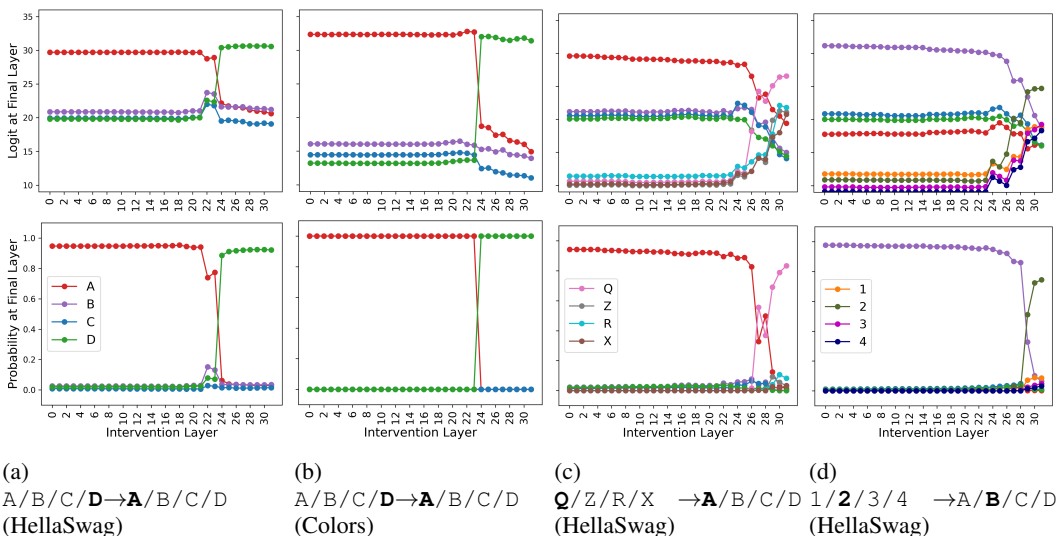

(a)
A/B/C/**D**→**A**/B/C/D
(HellaSwag)

(b)
A/B/C/**D**→**A**/B/C/D
(Colors)

(c)
**Q**/Z/R/X →**A**/B/C/D
(HellaSwag)

(d)
1/**2**/3/4 →A/**B**/C/D
(HellaSwag)

Figure 3: Average effect (top: logits; bottom: probits) of patching individual output hidden states for Olmo 7B 0724 Instruct ($x_B \rightarrow x_A$) on predictions correct under both prompts. Patterns are largely similar regardless of which position is used for replacement and the direction of replacement. See Fig. 16 for additional results.

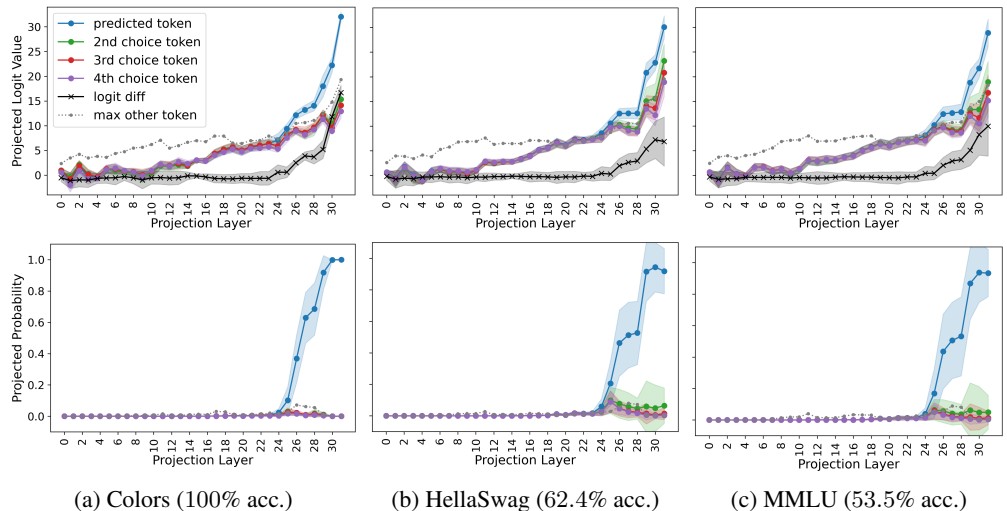

(a) Colors (100% acc.)    (b) HellaSwag (62.4% acc.)    (c) MMLU (53.5% acc.)

Figure 4: Average projected logits (top) and probits (bottom) of answer tokens at each layer for Olmo 0724 7B Instruct, for correct 3-shot predictions with the prompt A/B/C/D. See Fig. 13 for Llama 3.1 8B Instruct and Fig. 14 for Qwen 2.5 1.5B Instruct. See Fig. 15 for 0-shot results.

of the tokens of interest in vocabulary space, which affects greedy generation; 2) allows us to compare processing patterns across different models, which often have different absolute logit scales, and 3) allows us to disentangle logit changes that are *specific* to the tokens of interest as opposed to general trends to the entire model vocabulary. We thus use both metrics because they provide complementary information.

### 4.4 INITIAL OBSERVATIONS

**Activation patching and vocabulary projection are complementary.** As elucidated in Fig. 3b, activation patching highlights the key causal role that layer 24 plays in encoding the answer choice that Olmo 7B Instruct will predict on Colors. While layers prior to 24 may promote or demote a

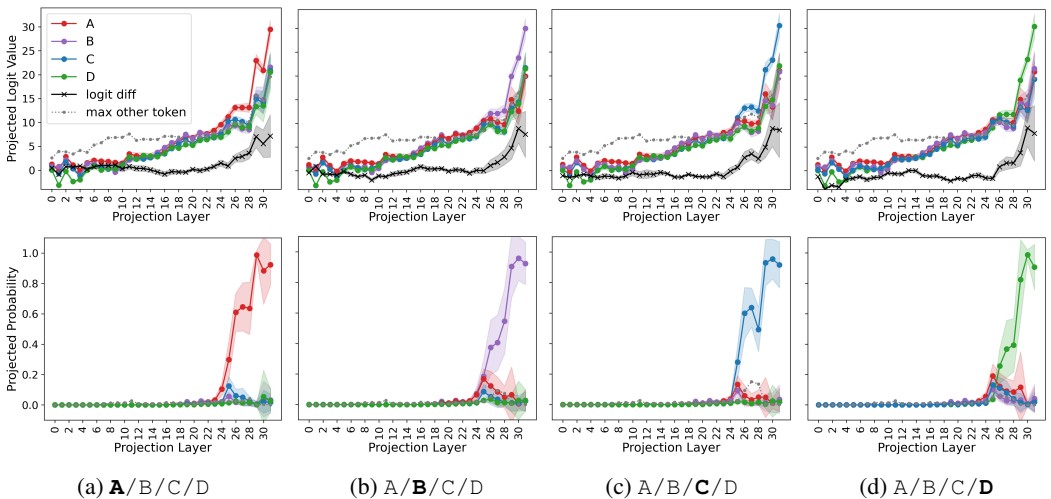

Figure 5: Average projected logits (top) and probits (bottom) of answer tokens at each layer for correct 3-shot predictions by Olmo 0724 7B Instruct on HellaSwag, with the prompt `A/B/C/D`. See Fig. 17 for Llama 3.1 8B Instruct, Fig. 18 for Qwen 2.5 1.5B Instruct, and Fig. 19 for 0-shot results.

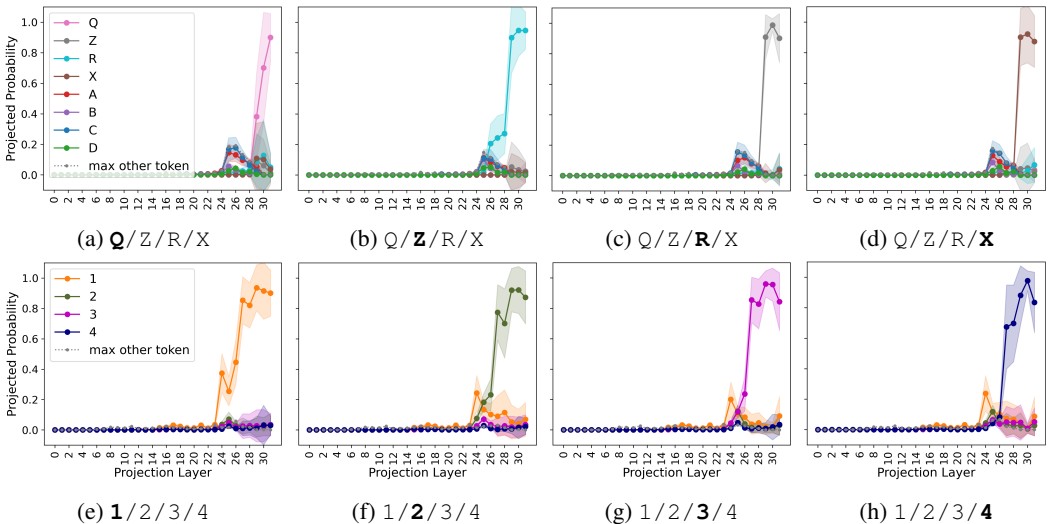

Figure 6: Average projected probits of answer tokens at each layer for correct 3-shot predictions by Olmo 0724 7B Instruct on HellaSwag for the `Q/Z/R/X` (top) and `1/2/3/4` (bottom) prompts with various correct answers (indicated in bold). Results for another random set of letters (`O/E/B/P`) are in Fig. 21; logit values are in Fig. 20. See Fig. 22 for Llama 3.1 8B Instruct and Fig. 23 for Qwen 2.5 1.5B Instruct.

specific token choice, the predicted token does not change, indicating that these layers' contributions to the residual stream are either nonexistent, or are overridden by layer 24's. Similarly, later layers appear to be carrying forward information contributed to the residual stream by layer 24 (while also slightly reducing the logits of the other answer choices), since they do not have any effect at *undoing* the change in predicted token caused by intervening on layer 24's output representation.

Compared with Fig. 3b, Fig. 4a illustrates the complementary nature of the two methods: activation patching reveals important mechanisms *before* hidden states are projectable to the vocabulary space, while vocabulary projection shows how relevant information in the residual stream appears in the vocabulary space over remaining layers. Indeed, one hypothesis that arises from these results is that it takes a couple of layers of processing for information encoded in a hidden state to become mapped to higher scores on those tokens.

## 5   ROLES OF SPECIFIC LAYERS

**Key Layers differ across models and # of shots, but are consistent across datasets and predicted tokens.**   Fig. 4 presents projected token scores for Olmo 7B Instruct across all three datasets, and Figs. 3a and 3b activation patches for two datasets. Despite Olmo 7B Instruct's varying performance on the datasets, key layers are largely consistent. Notably, layer 24 also plays the key causal role in the promotion of answer symbol tokens for HellaSwag, though layers 22-23 also have a small effect. In Fig. 4, the outputs of layers 26 and 29 lead to the largest logit and probit increases for the predicted answer token, with layer 31 serving to boost the logits of all tokens (this has no effect in probability space). Similar results are found for the Llama and Qwen models (Figs. 13 and 14), corroborating the hypothesis that the same mechanisms are responsible for promoting answer symbol letters across tasks, regardless of task complexity. However, comparing these graphs with Fig. 4 reveals that logit magnitudes and key layers differ across models, even for those of comparable size.

Comparing Fig. 4 (3-shot) to 0-shot results (Fig. 15), the role of in-context examples primarily manifests in not only increasing the logit values of all answer choices in the last layers of the model (29-31), but also leading to a substantial increase in the predicted token's value such that it far surpasses other tokens in the vocabulary and accumulates nearly all the probability mass. This does not happen in the 0-shot case despite overall accuracy being similar, and also coincides with a noticeable (temporary) demotion of valid answer symbols at layer 28.

We break down Fig. 4b by predicted answer letter (Fig. 5), observing that trends are generally similar across A, B, C, and D for Olmo 7B Instruct and Llama 3.1 8B Instruct (Fig. 17).[8] Qwen 2.5 1.5B exhibits a fairly strong early preference for A at layer 20 which may be due to the token's corpus frequency, but is otherwise similar across symbols (Fig. 18), particularly from layer 22 onward.

**Some answer symbols are produced in a two-stage process.**   Processing trends are noticeably different for some models when using a different symbol space (Q/Z/R/X, 1/2/3/4, or another set of random letter symbols, O/E/B/P). For random letters, Olmo and Qwen models first assign non-negligible probability to labels that are more likely or expected (A, B, C, D) even though they are not included in the prompt, before making an abrupt switch to the correct symbols (Q/Z/R/X or O/E/B/P) at layers 29 and 23, respectively (Fig. 6, top, Fig. 21 and Fig. 23). Activation patching reveals the same trend (Figs. 3c and 16). For example, for the Olmo model, patching in from prompts containing Q/Z/R/X or O/E/B/P only decisively promotes the new symbol from layer 29, noticeably later than layer 24 in the A/B/C/D experiments (Figs. 3a and 3b). This provides evidence that Olmo 7B Instruct solidifies final label predictions for "OOD" prompts in later layers, and could explain why some models struggle with OOD formats. In the case of a more standard prompt, 1/2/3/4, scores assigned to A/B/C/D remain negligible (Fig. 6, bottom).[9]

---

[8]We observe similar results in activation patching experiments to Figs. 3a and 3b when using different correct answer letters, further corroborating this trend.

[9]It is of note that activation patching from A/B/C/D to 1/2/3/4 (Fig. 3d) also identifies layer 29 as a key layer. This differs from Fig. 3a, where position but not symbols changed. Since correct answer position does not change in this experiment, one possible explanation is that layer 24 is causally attributed to encoding positional information, while layer 29 is attributed to encoding the relevant answer token at that position.

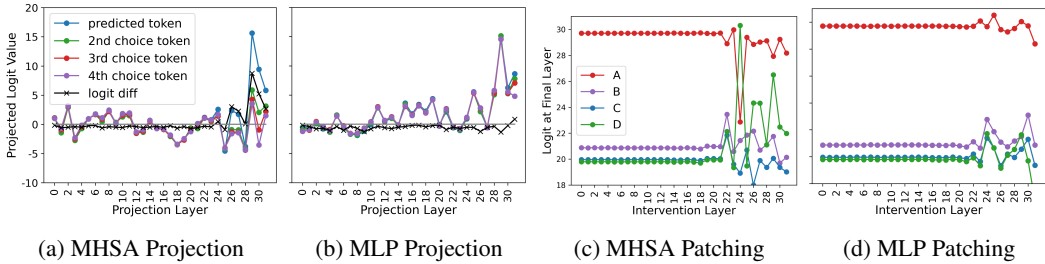

|  |  |  |  |
|---|---|---|---|
| (a) MHSA Projection | (b) MLP Projection | (c) MHSA Patching | (d) MLP Patching |

Figure 7: Avg. projected logits with the `A`/`B`/`C`/`D` and avg. logit values at the final layer (patching `A`/`B`/`C`/**`D`** →**`A`**/`B`/`C`/`D`) of hidden states output by different functions of the Olmo 7B 0724 Instruct model on HellaSwag. These are function-wise breakdowns of Figs. 3a and 4b. See Figure 12 for a breakdown of Figure 7c by attention heads.

## 6 ROLES OF LAYER COMPONENTS

We next investigate the role that MHSA and MLP functions play in the observed behavior.[10] weighted by their respective rows of the MHSA output matrix, since the MHSA output vector is a weighted sum of the heads' output vectors. See Eqs. (1) to (3) in Appendix A.2 for the derivation.

**Self-attention mechanisms dominate the production of answer choice symbols (over MLPs).** In Figs. 7a and 7b, we directly project the hidden states output by these functions (positions A and B in Fig. 10). Both functions map to large increases in logit values at layer 29. However, MLP outputs serve to increase the logit value of *all* answer choices, not to be discriminative (as noted by the near-0 logit difference line). Increases in the logit difference are attributed to the MHSA functions at layers 26, 27, and 29. This is despite the fact that MHSA functions only contain ˜half the learnable parameters of MLPs (˜67M vs. ˜135M per layer). However, attention heads have access to global context which an MLP does not have, making them especially useful for resolving back-references to portions of the input sequence or copying information from one position to another.

**Answer symbol production is driven by a sparse portion of the network.** The corresponding breakdown in the logit and probit differences from projecting individual attention head output vectors to the vocabulary space (Fig. 8a) shows that this effect is quite sparse, with 1-4 attention heads per layer (out of 32) projecting to non-negligible values on *any* answer choice symbol. We additionally observe high absolute magnitude overlap between Fig. 8a and Fig. 8b/8c, indicating that the same components promoting probability on *any* answer choices (Fig. 8a) are also generally *discriminative* in preferring one choice over the other (Fig. 8b/8c). However, the differences between Figs. 8b and 8c reveal that some heads play letter-specific roles. Similar trends hold for the Qwen model (Fig. 25). Activation patching results (Figs. 7c and 7d) further elucidate that the MHSA mechanism is driving the encoding of relevant information to predict the answer primarily at layer 24, with Fig. 12 demonstrating that a *single attention head* is responsible for this effect.

## 7 WHERE ARE POORLY-PERFORMING MODELS GOING WRONG?

**Our synthetic task separates formatted MCQA performance from dataset-specific performance.** In Fig. 9, our synthetic task confirms that there is a crucial point at which formatted MCQA skill is learned for Olmo 0724 7B base – between 80k and 100k training steps. Performance goes from near-random to near-100%, even for checkpoints with poor performance on more challenging datasets. This result highlights the value of a synthetic task: it helps to disentangle to what extent poor MCQA performance on a dataset is due to a lack of dataset-specific knowledge vs. an inability to perform formatted MCQA.

---

[10]Vocabulary projections in logit space of a layer's MLP and MHSA functions are a direct additive decomposition of the projections on the layer's final hidden state. This is also true for individual attention heads

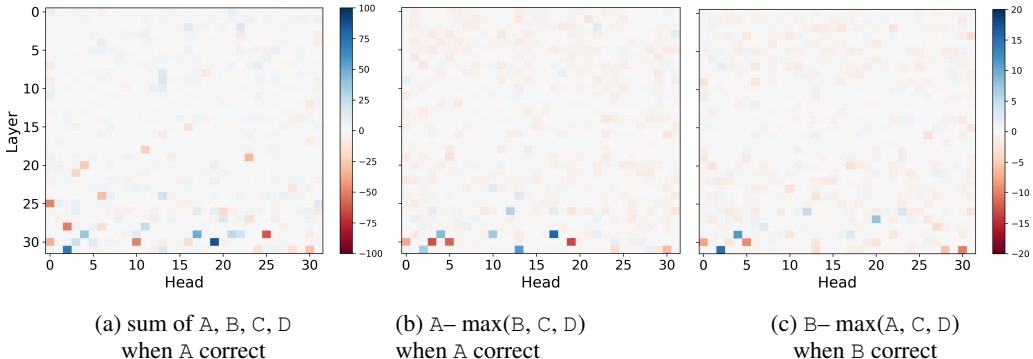

(a) sum of A, B, C, D
when A correct

(b) A− max(B, C, D)
when A correct

(c) B− max(A, C, D)
when B correct

Figure 8: Average **sum** (left) vs. **difference** (right two; separated by predicted letter) of logits when individual attention heads are projected to vocabulary space (Footnote 10). Plotted are results for correctly-predicted HellaSwag instances by Olmo 0724 7B Instruct. See Fig. 24 for probits. See Fig. 25 for Qwen 2.5 1.5B Instruct.

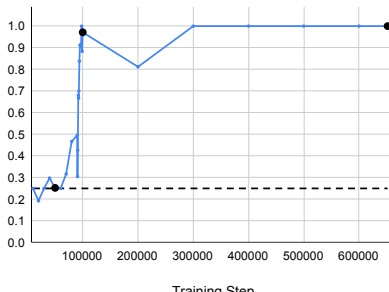

Training Step

Figure 9: 3-shot accuracy of various Olmo 0724 7B Base checkpoints on the Colors task. Avg. performance across all four correct answer positions is plotted, with the dotted line indicating random performance. MCQA format is learned at a specific (early) point in training. Black dots indicate checkpoints used in subsequent analysis, representing 3 distinct points on the learning curve. See Fig. 26 for 0-shot results.

**Poorly performing models cannot separate answer choice symbols in vocabulary space.** We use vocabulary projection to inspect differences between these checkpoints on the Colors task. Fig. 27 (Appendix) illustrates the separability between the answer choice symbols, which grows as a function of training for more steps. While at $100k$ and $200k$ steps the model represents a meaningful ranking of the answer tokens that results in high accuracy using the multiple-choice prediction rule, only with sustained training does the model widen the logit difference substantially, leading to the model assigning high probability to the predicted answer alone. The small differences in logit scores observed at earlier checkpoints may explain why small edits to network parameters are effective at decreasing label bias (Li & Gao, 2024).

Finally, we observe in Fig. 28 (Appendix) that the ability to do symbol binding at the final training checkpoint, as demonstrated on Colors, does not alleviate small logit differences between answer symbols, even for correct predictions, on datasets on which the model is less performant: scores assigned to correct answers are both higher variance and tempered in layers 30-31 in these cases.

## 8  CONCLUSION

This work provides novel insights into the mechanisms language models employ when answering formatted multiple-choice questions. Our analysis leverages two complementary techniques: vocabulary projection and activation patching. Using carefully designed prompts and a synthetic task, we uncover a series of stages, attributable to a sparse set of specialized network components, in which models select, promote, and align (for OOD prompts) the answer choice symbol they predict. With MCQA datasets such as MMLU routinely used as a performance signal for model development, it is important to disentangle and test formatted MCQA ability in order to build models that operate in more robust and reliable ways. Our work makes an important first step towards this goal.

## REPRODUCIBILITY STATEMENT

To the best of our ability, we have included all details necessary to reproduce our experiments in the text of this paper and accompanying Appendix. We have also open-sourced our code (`https://github.com/allenai/understanding_mcqa`) and Colors dataset (`https://huggingface.co/datasets/sarahwie/copycolors_mcqa`). We fixed random seeds to ensure full reproducibility of our experiments.

## ACKNOWLEDGMENTS

We thank the anonymous reviewers, members of the Aristo team at AI2, and members of the H2Lab at the University of Washington for valuable feedback. YB was supported by the Israel Science Foundation (grant No. 448/20), an Azrieli Foundation Early Career Faculty Fellowship, and an AI Alignment grant from Open Philanthropy. This research was partially funded by the European Union (ERC, Control-LM, 101165402). Views and opinions expressed are however those of the author(s) only and do not necessarily reflect those of the European Union or the European Research Council Executive Agency. Neither the European Union nor the granting authority can be held responsible for them.

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

# A  APPENDIX

## A.1  DATASET DETAILS

**HellaSwag** (Zellers et al., 2019) is a 4-way multiple-choice commonsense natural language inference dataset. The goal is to select the best completion for a given sentence prompt. We sample a fixed set of 1000 instances from the test set used in our experiments. We sample 3 random training set instances to serve as in-context examples.

**MMLU** (Hendrycks et al., 2021), or the "Massive Multitask Language Understanding" benchmark, spans 57 different topical areas. The questions are 4-way multiple-choice spanning subjects in social sciences, STEM, and humanities that were manually scraped from practice materials available online for exams such as the GRE and the U.S. Medical Licensing Exam. We sample a fixed set of 1000 instances from the test set used in our experiments. We sample a fixed set of 3 in-context example instances from the 5 provided for each topical area.

**In-context example selection.**   We randomly select the correct answer position for each in-context example (which comes to positions 012; or A B C, Q R X, or 1 2 3, depending on the prompt format. We find that models have more consistent performance when in-context examples are used (particularly, the models we benchmark which are not instruction-tuned), but we do also include 0-shot results for a subset of strong models, where the inclusion of in-context examples does not have a large effect on performance (Fig. 11).

## A.2  TRANSFORMER BACKGROUND

The output hidden states of transformer models (Vaswani et al., 2017) are a linear combination of the outputs of non-linear functions (i.e., multi-head self-attention and multi-layer perceptrons) at each layer. For most models, which sequentially apply multi-head self-attention (MHSA) and a multi-layer perceptron (MLP), for input hidden state $\mathbf{x}_{\ell-1} \in \mathbb{R}^d$, the output hidden state $\mathbf{x}_\ell \in \mathbb{R}^d$ of layer $\ell$ is defined by the following equation:

$$\mathbf{x}_\ell = \mathbf{x}_{\ell-1} + \text{MHSA}_{\theta_\ell}\big(\text{LN}(\mathbf{x}_{\ell-1})\big) + \text{MLP}_{\theta_\ell}\Big(\mathbf{x}_{\ell-1} + \text{MHSA}_{\theta_\ell}\big(\text{LN}(\mathbf{x}_{\ell-1})\big)\Big) \qquad (1)$$

Where $\text{MHSA}_{\theta_\ell}$ and $\text{MLP}_{\theta_\ell}$ represent the multi-head self-attention operation and multi-layer perceptron, respectively, defined by their unique parameters at layer $\ell$, and LN represents layer normalization.[11] Note that $+$ represents simple vector addition, which serves to establish residual connections. This is possible because the output of each operation (MHSA, MLP, and LN) is a vector $\in \mathbb{R}^d$. We visualize a single layer's block structure in Fig. 10.

Elhage et al. (2021) hypothesize and provide evidence for the "residual stream" view of Transformer inference, in which MHSA and MLP functions "read from" and "write to" the residual stream, which carries information through the layers. Additional evidence that MHSA and MLP functions promote specific tokens by writing to the residual stream is given by Geva et al. (2022b).

Given an input token embedding $\mathbf{x}_0 \in \mathbb{R}^d$ and applying Eq. (1) recursively, the final hidden state of a Transformer with $L$ layers resolves to:

$$\mathbf{x}_L = \mathbf{x}_0 + \sum_{\ell=0}^{L-1}\left[\text{MHSA}_{\theta_{\ell+1}}\big(\text{LN}(\mathbf{x}_\ell)\big) + \text{MLP}_{\theta_{\ell+1}}\Big(\mathbf{x}_\ell + \text{MHSA}_{\theta_{\ell+1}}\big(\text{LN}(\mathbf{x}_\ell)\big)\Big)\right] \quad (2)$$

The output of a MHSA function can be further broken down into a sum of the output of each attention head. For the input vector to layer $\ell$, $\mathbf{x}_{\ell-1}$, the output of MHSA is computed as the concatenation of each head's vector output, $\text{Att}_h^{(\ell)}\big(\text{LN}(\mathbf{x}_{\ell-1})\big) \in \mathbb{R}^{d/H}$, times an output weight matrix $W_O^{(\ell)} \in \mathbb{R}^{d \times d}$ which can be simplified into a sum as follows (originally elucidated in (Elhage et al., 2021)):

$$\text{MHSA}_{\theta_\ell}\big(\text{LN}(\mathbf{x}_{\ell-1})\big) = \sum_{h=1}^{H} W_{O,h}^{(\ell)} \cdot \text{Att}_h^{(\ell)}\big(\text{LN}(\mathbf{x}_{\ell-1})\big) \quad (3)$$

where $W_{O,h}^{(\ell)} \in \mathbb{R}^{d \times (d/H)}$ are the specific columns of $W_O^{(\ell)}$ corresponding to head $h$. When we perform experiments on individual attention heads, we are referring to the individual components of this sum (i.e., weighted attention head outputs).

# B  STRENGTHS AND WEAKNESSES OF VOCABULARY PROJECTION VS. ACTIVATION PATCHING

Vocabulary projection provides a meaningful notion of how hidden states assign probabilities to tokens in the vocabulary space defined by the unembedding matrix, but it is not a causal intervention, and it cannot uncover ways in which hidden states could be working to promote tokens in other linear (or non-linearly decodable) subspaces. For these reasons, negative results in earlier layers are uninformative. We supplement our findings by using activation patching to localize effects at earlier layers.

Activation patching makes some assumptions about the independence of model components to make computation tractable (namely, that one can patch in individual hidden states in isolation to measure their effect on the network as opposed to patching all possible combinations of states). A nascent line of work focuses on making activation patching more efficient so that a larger number of states and/or state combinations can be intervened on Kramár et al. (2024); Syed et al. (2024); this is an interesting direction for future work.

---

[11]$\text{MHSA}_{\theta_\ell}$ also receives as input the hidden states at layer $\ell - 1$ from other token positions in the rollout (as dictated by the attention mask), which we omit here for brevity.

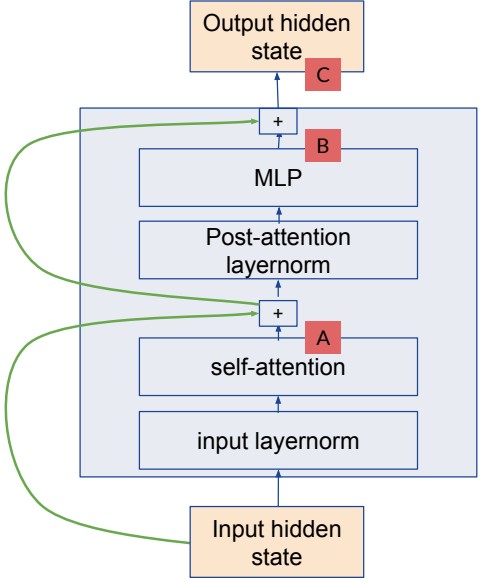

Figure 10: Structure of a single layer in the Transformer architecture for the models we study. Green lines indicate residual connections. We perform vocabulary projection and activation patching on different representations both pre-and-post residual combination, indicated by the letters.

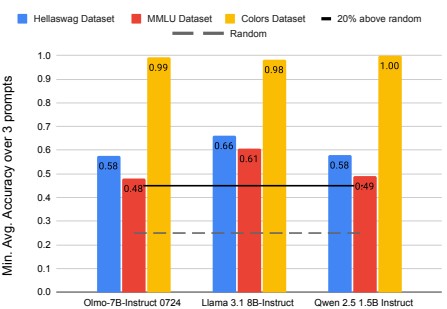

Figure 11: 0-shot performance results on Colors, Hellaswag and MMLU datasets. Plotted is the minimum accuracy across A/B/C/D, Q/Z/R/X, and 1/2/3/4 prompts, where the accuracy for each prompt is taken as the average over the correct answer choice being at each position.

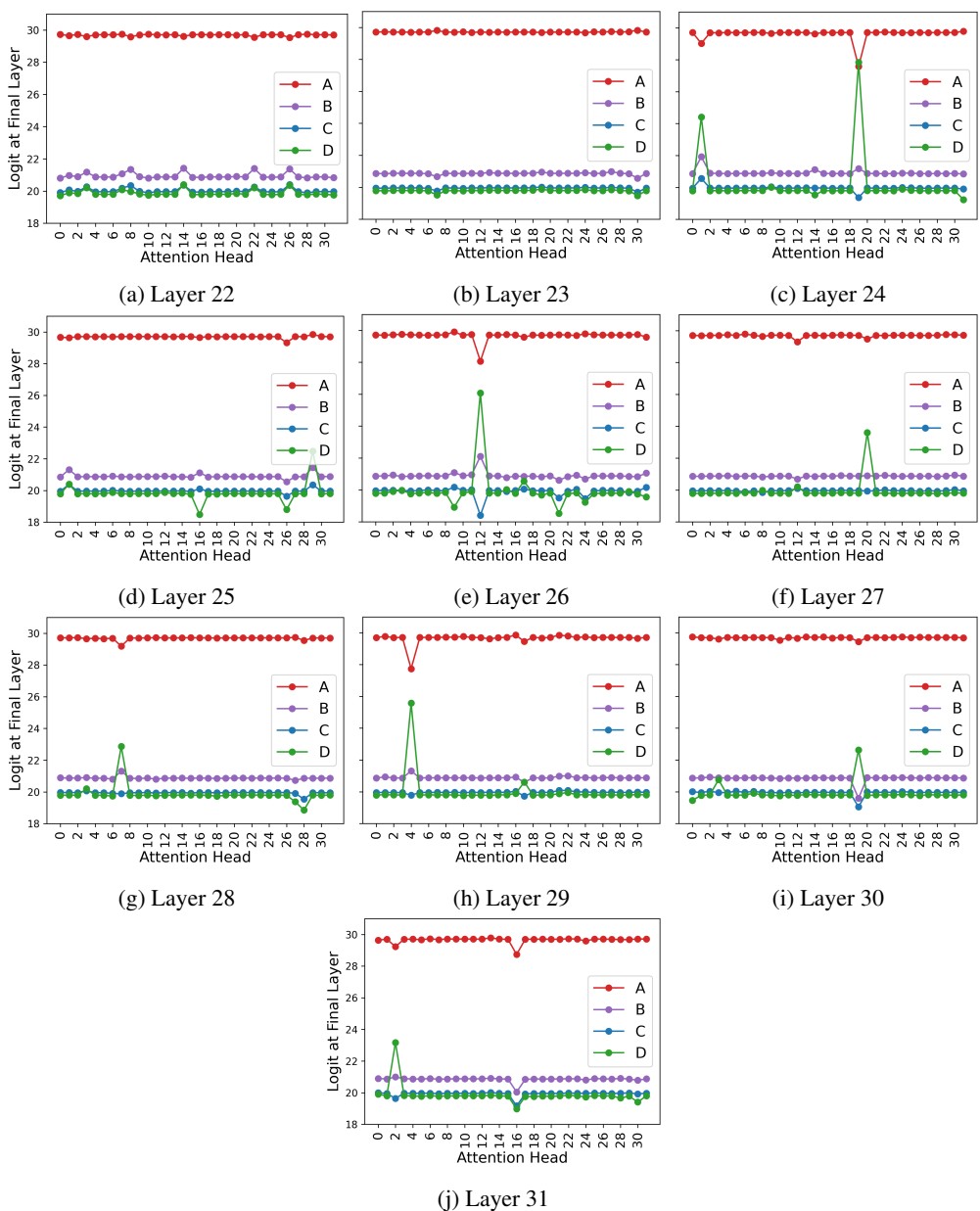

Figure 12: Logit values at the final layer (patching `A`/`B`/`C`/**`D`** →**`A`**/`B`/`C`/`D`) for each attention head in the last 10 layers of the Olmo 7B 0724 Instruct model on a subset of HellaSwag. We do not patch attention heads before layer 22 because no effect is observed in Figure 7c. The results demonstrate that the promotion of specific answer choices (i.e., the spikes in Figure 7c) are attributed to specific heads per layer, demonstrating the unique causal roles of individual heads.

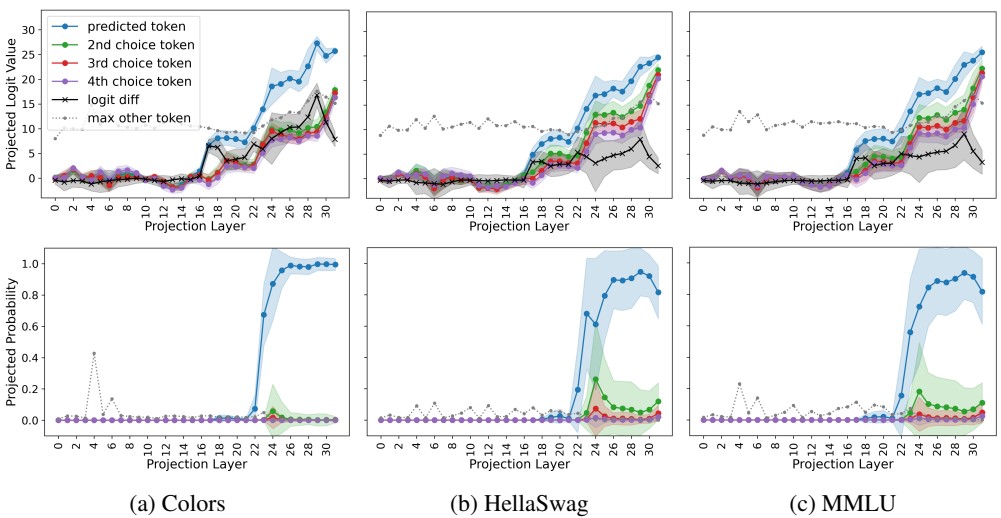

Figure 13: Fig. 4 results on Llama 3.1 8B Instruct.

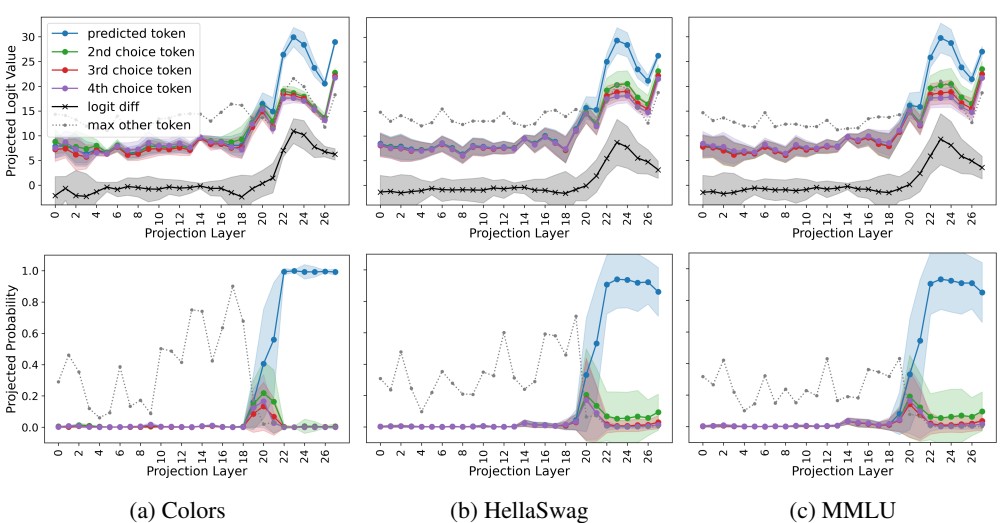

Figure 14: Fig. 4 results on Qwen 2.5 1.5B Instruct.

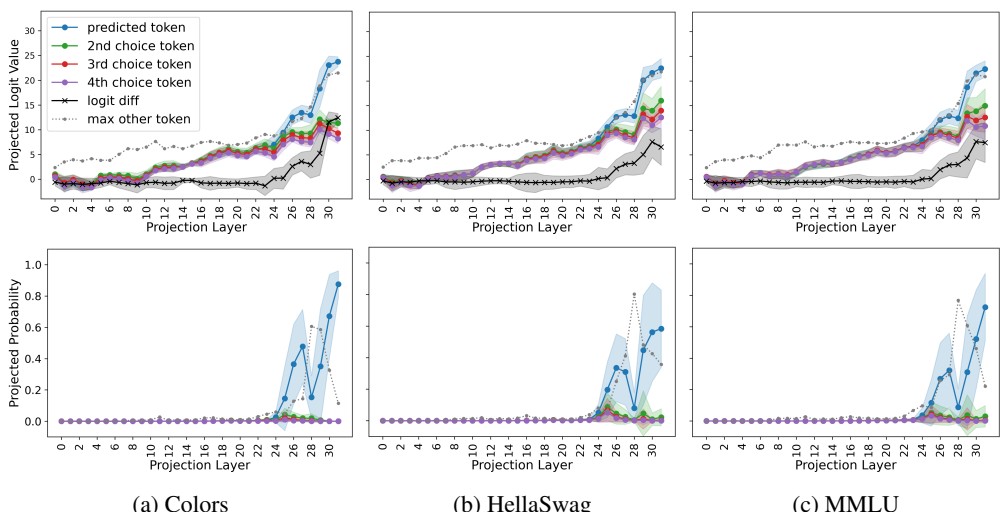

(a) Colors       (b) HellaSwag       (c) MMLU

Figure 15: 0-shot version of Fig. 4.

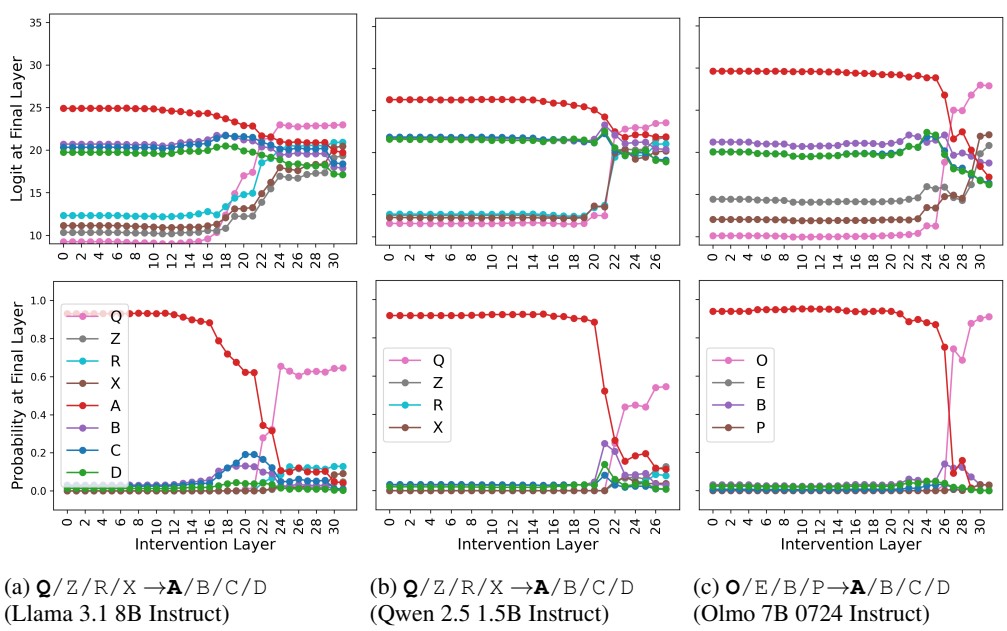

(a) **Q**/Z/R/X →**A**/B/C/D
(Llama 3.1 8B Instruct)

(b) **Q**/Z/R/X →**A**/B/C/D
(Qwen 2.5 1.5B Instruct)

(c) **O**/E/B/P→**A**/B/C/D
(Olmo 7B 0724 Instruct)

Figure 16: Average effect (top: logits; bottom: probits) of patching individual output hidden states for various models on predictions correct under both prompts on the **Hellaswag** dataset. See Fig. 3 for more details.

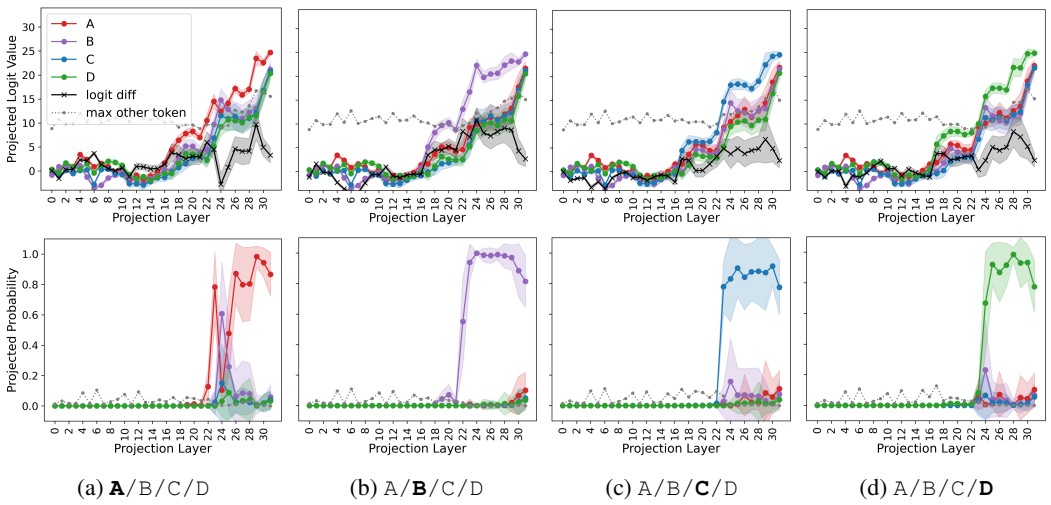

Figure 17: Fig. 5 for Llama 3.1 8B Instruct.

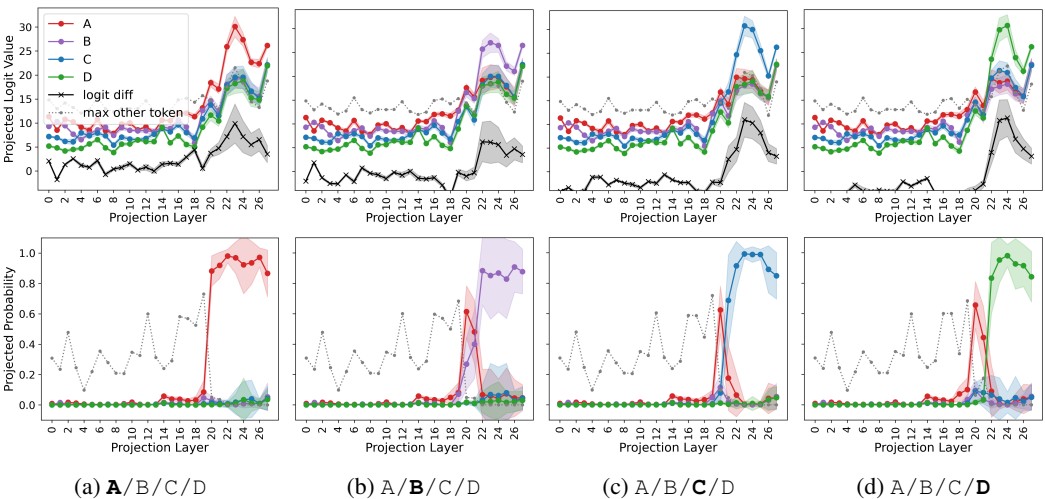

Figure 18: Fig. 5 for Qwen 2.5 1.5B Instruct.

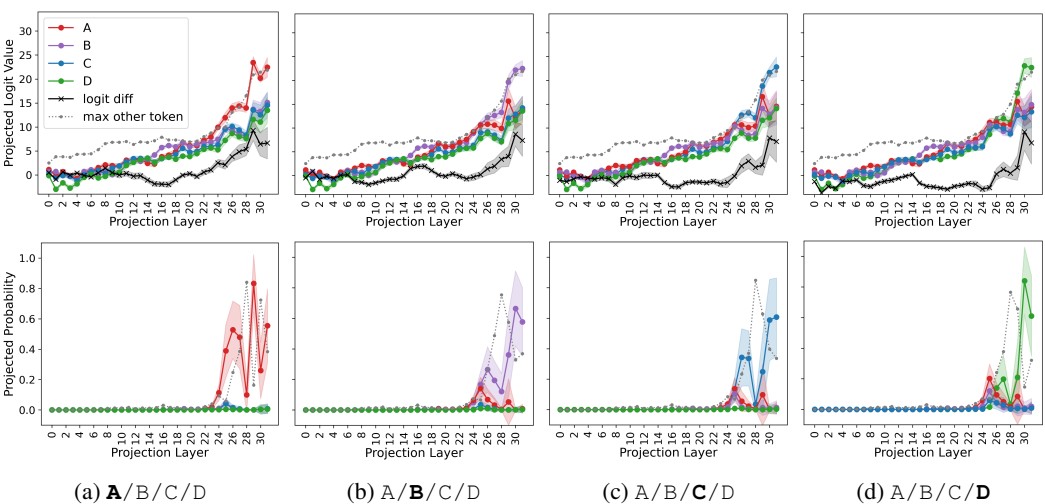

Figure 19: 0-shot version of Fig. 5.

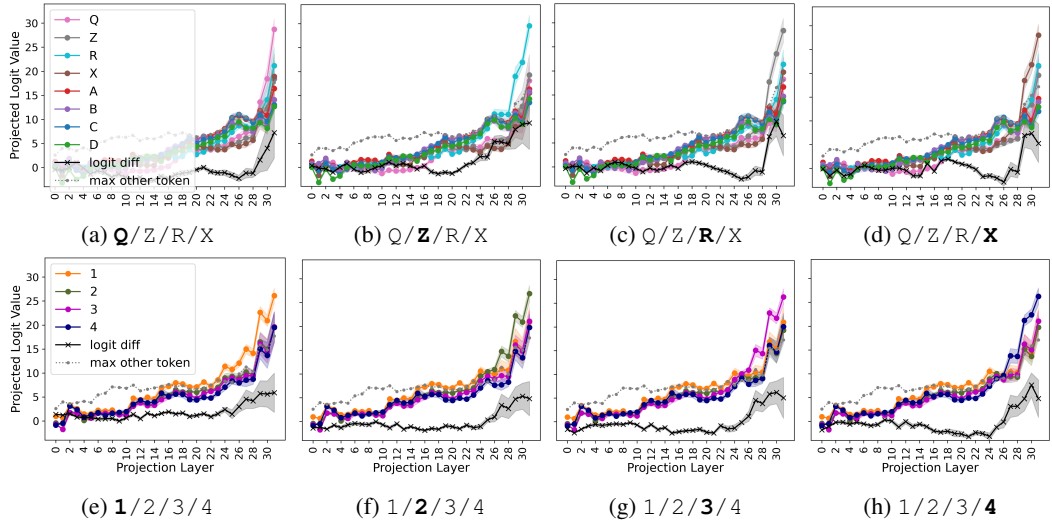

Figure 20: Logit plots for Fig. 6. See Fig. 22 (top) for Llama 3.1 8B Instruct and Fig. 23 (top) for Qwen 2.5 1.5B Instruct.

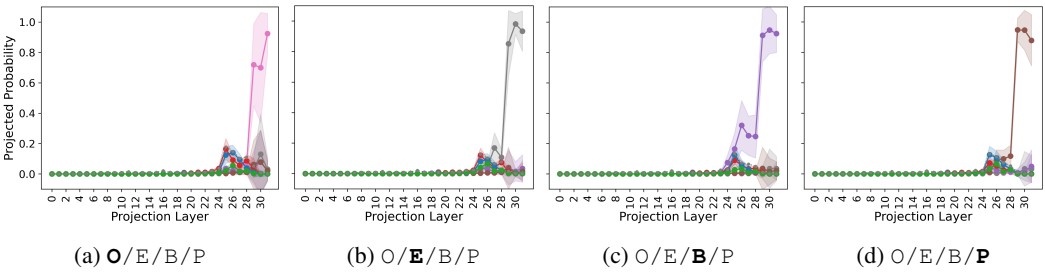

Figure 21: Average projected probits of answer tokens at each layer for correct 3-shot predictions by Olmo 7B 0724 Instruct for the O/E/B/P prompt with various correct answers (indicated in bold).

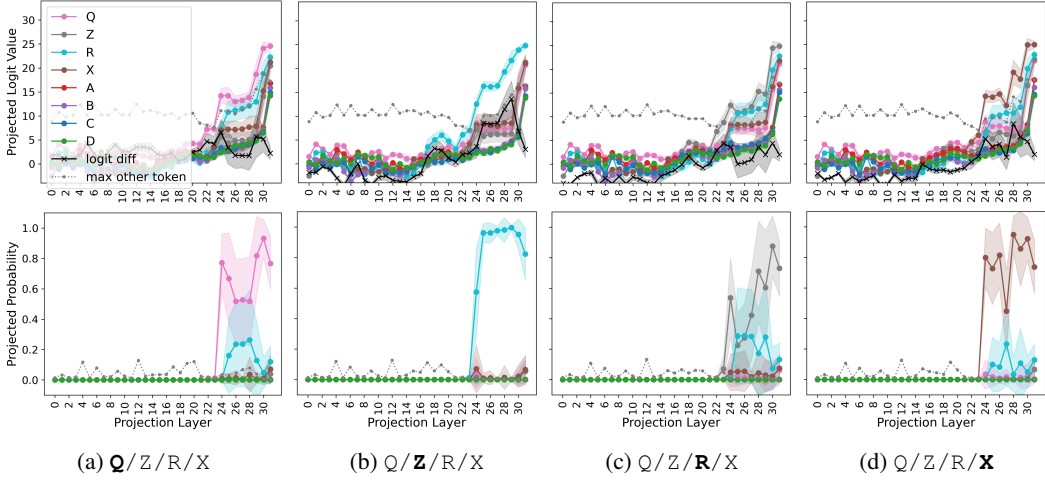

Figure 22: Average projected logits (top) and probits (bottom) of answer tokens at each layer for correct 3-shot predictions by Llama 3.1 8B Instruct for the Q/Z/R/X prompt with various correct answers (indicated in bold).

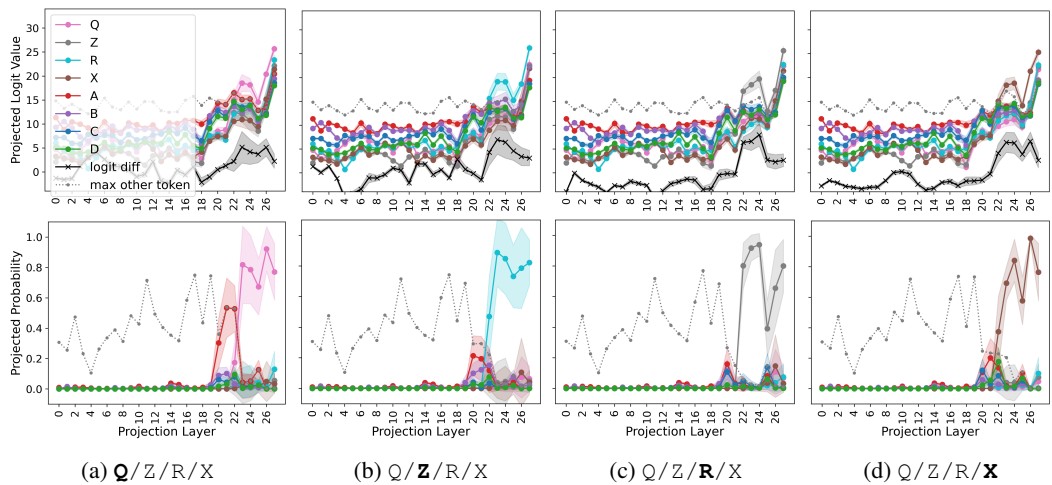

Figure 23: Average projected logits (top) and probits (bottom) of answer tokens at each layer for correct 3-shot predictions by Qwen 2.5 1.5B Instruct for the Q/Z/R/X prompt with various correct answers (indicated in bold).

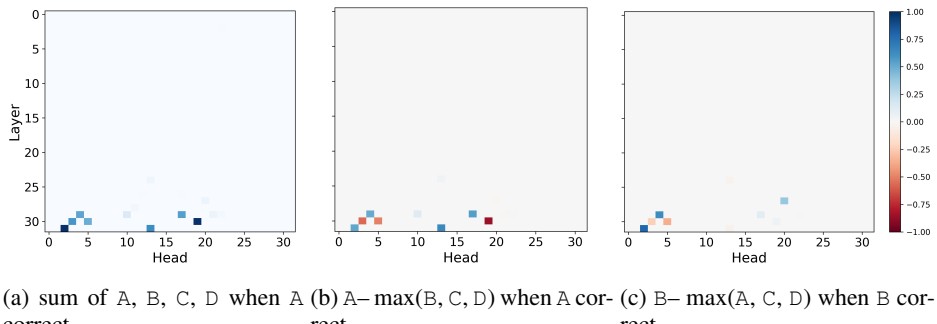

(a) sum of A, B, C, D when A correct
(b) A− max(B, C, D) when A correct
(c) B− max(A, C, D) when B correct

Figure 24: Probit plot version of Fig. 8. See Fig. 25 for Qwen 2.5 1.5B Instruct.

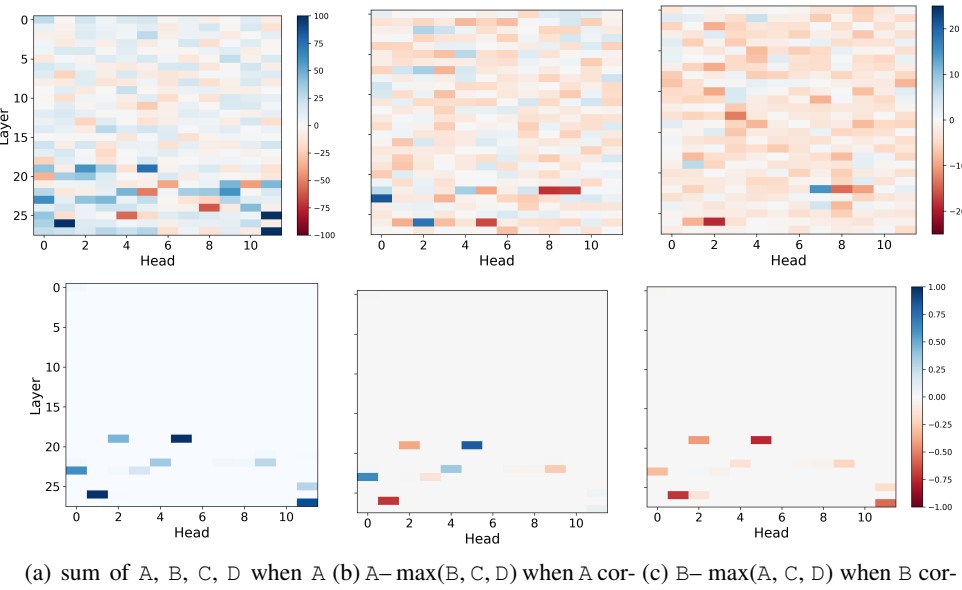

(a) sum of A, B, C, D when A correct
(b) A− max(B, C, D) when A correct
(c) B− max(A, C, D) when B correct

Figure 25: Fig. 8 for Qwen 2.5 1.5B Instruct.

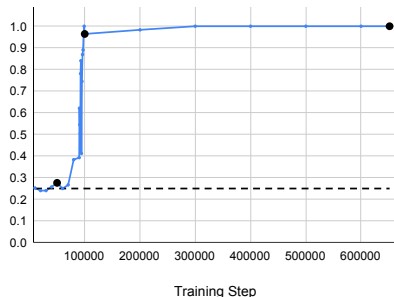

Figure 26: 0-shot accuracy of various Olmo 0724 7B Base checkpoints on the Colors task. See Fig. 9 for more info.

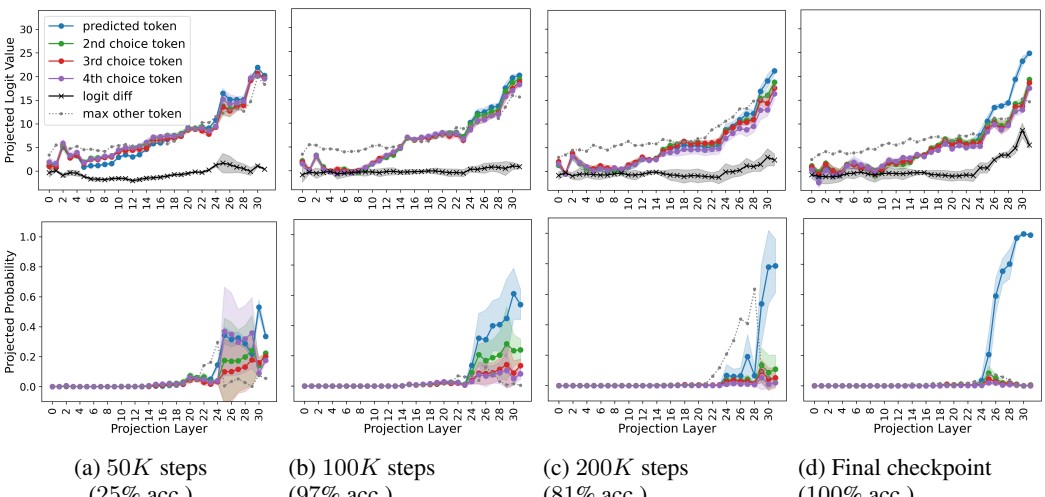

(a) $50K$ steps (25% acc.)

(b) $100K$ steps (97% acc.)

(c) $200K$ steps (81% acc.)

(d) Final checkpoint (100% acc.)

Figure 27: Average projected logits (top) and probits (bottom) of answer tokens at the final token position across each layer for correctly-predicted instances from the Prototypical Colors dataset with the 3-shot A/B/C/D prompt, across various Olmo 0724 7B Base checkpoints. The final checkpoint is at approx. $652K$ steps.

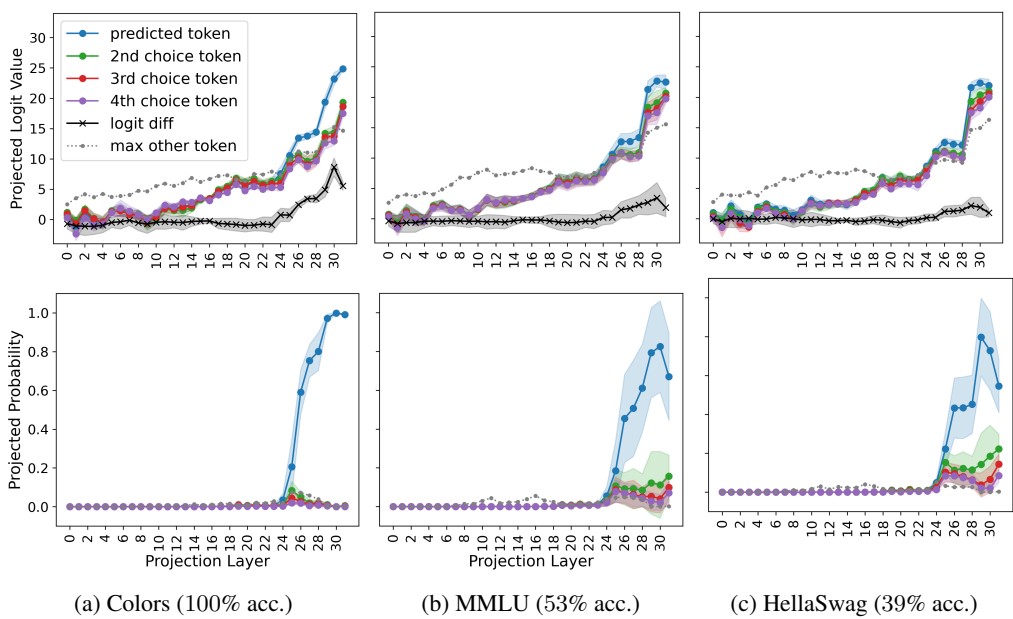

(a) Colors (100% acc.)    (b) MMLU (53% acc.)    (c) HellaSwag (39% acc.)

Figure 28: Average projected logits (top) and probits (bottom) of answer tokens at the final token position across each layer for correctly-predicted instances for Olmo 0724 7B Base with the 3-shot `A/B/C/D` prompt, across all 3 datasets.

