# OpenReview forum: "Answer, Assemble, Ace: Understanding How LMs Answer Multiple Choice Questions"
_ICLR.cc/2025/Conference — ICLR 2025 Spotlight_

### Official Review · Reviewer_WzgP · 2024-10-19

**Soundness:** 4
**Presentation:** 4
**Contribution:** 3
**Rating:** 8
**Confidence:** 3

**Summary:**

The authors study the mechanisms that exist in LLMs that enable them to successfully answer multiple choice questions. They primarily focus on three models from different model families, chosen for their well-above-average accuracy across answer orderings and datasets. They use in their study three datasets: MMLU, Hellaswag, and a synthetic dataset ("Colors"). Mechanism study is done with activation patching and vocabulary projection. Among the authors' key findings are:
* There is a specific layer or layers that are responsible for promoting the letter to be chosen.
* Letter selection is handled by the multi-head self-attention part of the model, and in particular by a small proportion of total heads.
* When unusual symbol groups (e.g., Q/Z/R/X) are used, the model first promotes A/B/C/D and then abruptly changes to promoting the unusual symbol group.
* Models that don't perform well on multiple choice cannot effectively separate the answer letters in vocabulary space.

**Strengths:**

This paper seems quite original. It is well contextualized with respect to prior work, and has novel contributions. The authors' claims are clear and are backed by broad and consistent experimental results. The interpretability findings may be of interest to researchers working in the space of MCQA and LLMs, or in the space of model interpretability in general.

**Weaknesses:**

* I'm surprised that the OLMo 0724 7B Instruct (Figure 9) study used 0-shot accuracy. In general, the authors use 3-shot accuracy in the main body of the paper, which seems prudent as the 0-shot case is somewhat ambiguous (it's unclear whether the correct response is answer label or answer text). It would be interesting to see 3-shot results for the study or to hear a justification for use of 0-shot accuracy.
* The findings in this paper rely on some assumptions inherent in activation patching and vocabulary projection. For example, that it's reasonable to project hidden layers from early in the network to the vocabulary space. I didn't take this potential weakness into account in my rating of the paper, as I'm not intimately acquainted with very recent mechanistic interpretability work. But it is something the authors could address in the paper if desired.

**Questions:**

* As mentioned in "Weaknesses" I'd be interested to hear the reason for using 0-shot accuracy in Figure 9.

---

> ### Author Response · Authors · 2024-11-22
> **Author Response**
>
> ### Thank you for your positive assessment of our paper!
>
> > *"...the OLMo 0724 7B Instruct (Figure 9) study used 0-shot accuracy…It would be interesting to see 3-shot results for the study or to hear a justification for use of 0-shot accuracy.”*
>
> Our primary motivation for showing the 0-shot learning curve is that it’s a lower bound on 3-shot performance, and we see in Fig. 9 that the model clearly learns the task (without needing in-context examples) early in training. But you raise a good point, and we’ll update Fig. 9 (and additionally, Figs. 21-22) to be 3-shot for consistency with the rest of the paper.
>
> > *"The findings in this paper rely on some assumptions inherent in activation patching and vocabulary projection…it is something the authors could address in the paper if desired.”*
>
> Thanks for pointing this out. This is exactly why we include both methods, because they are complementary (lines 238-240). We initially had more discussion of their strengths & weaknesses that got cut for space– we will add this back in the Appendix of the updated PDF (coming shortly).

---

### Official Review · Reviewer_xW8X · 2024-10-29

**Soundness:** 3
**Presentation:** 3
**Contribution:** 3
**Rating:** 6
**Confidence:** 4

**Summary:**

This research work studies how LLMs solve MCQA task. Authors show which layers and layer components play the most crucial role in this process and at which point the OLMO model starts to learn to solve this task. They also find out how different is this process when the options are named by other symbols (not A,B,C,D).

**Strengths:**

- The paper analyses an important topic, since MCQA tasks are very common in LLM benchmarks.
- Authors came up with a good idea for their synthetic dataset. That dataset allows them to distinguish a model's understanding of MCQA format from it's knowledge in particular area, that is important for proper analysis.
- A number of experiments are done on several models: Olmo, LLaMA 3.1, Qwen: -chat and -base versions. These models look like an appropriate objects for this study.
- At the second half of their paper, authors especially concentrate on those models that are good in solving three MCQA tasks (synthetic, HellaSwag and MMLU). It is also a good move, because it allows to ignore "noisy" properties of weak models that could mislead researcher and reader otherwise.
- Authors use a good practices in MCQA research, i.e. they permute A/B/C/D symbols and consider alternative symbols for answer options.

**Weaknesses:**

- The claim in the lines 518-519 (_"these results demonstrate that an inability to disentangle answer symbol tokens in vocabulary space is a property of poorly-performing models"_) is not sufficiently supported. Such claim definitely cannot be deduced from the analysis of the checkpoints of only one particular Olmo 0724 7B base model (however, at lines 79-81 you make much more humble version of this claim, with which I agree).
- In the lines 76-78 you say: _"We discover that the model’s hidden states initially assign high logit values to expected answer symbols (here, A/B/C/D) before switching to the symbols given in the prompt (here, the random letters Q/Z/R/X)."_, but this claim is also not sufficiently supported. You only showed this effect for one non-standard letter set, and, again, I see only Olmo 7B experiments here (please, correct me if I'm wrong).
- Some parts of the paper are poorly-written and unclear. E.g. I don't understand the caption for the Figure 1, especially the sentence _"Finally, when we switch to more unusual answer choice symbols, one behavior by which models adjust is to initially operate in the space of more standard tokens, such as A/B/C/D, before aligning probability to the correct symbols at a late layer"_. See also "Question" sections for the questions about other Figures.
- There are following minor problems:
-- Text in the diagrams is extremely small. Please, make it larger to make it readable.
-- The paper contains typos such as word "projectioan" on line 066. Please, check the grammar of your text.
-- Citations in the line 264 should be put inside the braces.

**UPDATE**: Authors addressed the second major weakness by adding the results for other letter sets and other models. Due to this, **I raised my main score from 5 to 6** and "Soundness" score from 2 to 3. They also answered the questions, made their points more clear and addressed minor problems in the text of the paper. Due to this I raised "Presentation" score from 2 to 3.

**Questions:**

- Fig.9: Do you have such curve for any other model, aside of Olmo 0724 7B base?
- Fig.4: I don't understand the purpose of making this figure. What should it show to us?
- Fig.8: Can you, please, elaborate the captions a), b), c)?
- In the most of your experiments, you only consider the models from the range of 1.5-8B (smaller models turned out to be too weak to make good MCQA predictions). Do you have a physical possibility to do at least some experiments with some bigger models (at least, 13B)? It would be especially interesting to see the effect of switching from A/B/C/D to Q/Z/R/X (and other non-standard letters/symbols sets) at such model.

---

> ### Author Response · Authors · 2024-11-22
> **Author Response**
>
> ### Thank you for acknowledging many positive aspects of our work; we can alleviate many of the weaknesses you mention with clarifications presented here and writing updates to our PDF. We've updated text in diagrams to be as large as possible given the page constraint, and made another pass to fix typos and put citations in brackets; thank you for these catches. Updated PDF coming soon!
>
> > *"The claim in the lines 518-519…cannot be ​​deduced from the analysis of the checkpoints of only one particular Olmo 0724 7B base model (however, at lines 79-81 you make much more humble version of this claim, with which I agree).”*
>
> Thank you for pointing this out; we will update this sentence in our new PDF (coming shortly) to instead reflect the more humble version of the claim.
>
> > *"In the lines 76-78…you only showed this effect for one non-standard letter set, and, again, I see only Olmo 7B experiments here (please, correct me if I'm wrong).”*
>
> When we say “the model’s hidden state”, we are referring to the Olmo 7B Instruct model mentioned earlier in the sentence. We are explicit in the paper that this result is for the Olmo model (lines 409 and 416), and thus represents one means by which models produce OOD letters (line 41). We'll further clarify this in the updated PDF though, and you're right that it will be valuable for us to add experiments for Llama and Qwen.
>
> As for other non-standard answer symbols, we did observe this effect consistently in our preliminary experiments, but did not include these results in the paper. This is valuable feedback and we will add results on another randomly-selected letter set to show that our findings are robust.
>
> ### Your questions:
>
> > *"I don't understand the caption for the Figure 1, especially the sentence "Finally, when we switch to more unusual answer choice symbols, one behavior by which models adjust is to initially operate in the space of more standard tokens, such as A/B/C/D, before aligning probability to the correct symbols at a late layer".”*
>
> This is referring to our finding in lines 76-78 and in Section 5 lines 407-419 under the heading “Some answer symbols are produced in a two-stage process”: that for the Olmo 7B Instruct model, hidden states initially assign high logit values to expected answer symbols (A/B/C/D) before switching to the symbols given in the prompt. We'll try to adjust the phrasing to make this clearer.
>
> > *"Fig.9: Do you have such curve for any other model, aside of Olmo 0724 7B base?”*
>
> No, and unfortunately we are unable to produce one. The Llama and Qwen developers did not release any intermediate model checkpoints, and the Olmo 1B model doesn’t do well enough on the synthetic task at the end of training to warrant constructing one (see Fig. 2).
>
> > *"Fig.4: I don't understand the purpose of making this figure. What should it show to us?”*
>
> The purpose of the figure is to show that trends in promoting answer choices in vocabulary space are largely similar across tasks, despite the fact that the difficulty and subject matter of the tasks vary substantially and model performance differs as a result (i.e., from left to right: 55%, 51%, and 100% accuracy). We elaborate on this in lines 392-394, but if additional clarification or discussion would be valuable, we are happy to add. We’ve also added the accuracies to the figure caption.
>
> > *"Fig.8: Can you, please, elaborate the captions a), b), c)?”*
>
> Plots b) and c): this is the difference in logits on the answer choice tokens produced by a hidden state projected to the vocabulary space (described in Section 4.2, specifically lines 254-255). This is equivalent to the “logit difference” lines in Figures 5a and 5b, except now at the more fine-grained level of individual attention heads instead of hidden state outputs. We plot this to show the extent to which attention heads specialize *by letter* (by comparing how the two plots differ). Additionally, while the model is ultimately producing positive values (blue) for the instances in both graphs that result in correct predictions, the plots show that not all attention heads contribute positively towards the model predicting the correct answers.
>
> Plot a): this is the sum of logit values for all answer choices (logit of A + logit of B + logit of C + logit of D) using the same vocabulary projection method. This is equivalent to summing the A, B, C, and D lines in Figure 5a, except now at the more fine-grained level of individual attention heads instead of hidden state outputs. We plot this to show the overlap between plots a), b) and c): many attention heads appear to play a role in both generating *any* valid answer choice symbol and outputting the *correct* symbol (lines 478-479).
>
> We will include additional context for these plots in the paper.

---

> > ### Author Response · Authors · 2024-11-22
> > **continued**
> >
> > > *“Do you have a physical possibility to do at least some experiments with some bigger models (at least, 13B)?”*
> >
> > The only other sizes of Llama 3.1 available are 70B and 405B, and Olmo does not have any model sizes apart from those we tested (1B and 7B). We included Qwen 2.5 0.5B and 1.5B sizes in the paper to provide a size contrast to the 7B and 8B models; we feel these 3 model families and 5 sizes are fairly representative.
> >
> > While we unfortunately did not have the resources to scale to super large (70B+) sizes, we note this is the case in virtually all current academic mechanistic interpretability research. However there are some promising up-and-coming initiatives (such as https://arxiv.org/abs/2407.14561) to allow for analysis at larger scales soon, so we hope we can scale in future work.

---

> > > ### Comment · Reviewer_xW8X · 2024-11-25
> > >
> > > Thank you for your response. Could you please confirm if any updates have been made to the paper?

---

> ### Author Response · Authors · 2024-11-27
> **Updated PDF**
>
> We have updated the PDF with the requested changes (see general response for the details).

---

### Official Review · Reviewer_FfQL · 2024-10-29

**Soundness:** 3
**Presentation:** 3
**Contribution:** 2
**Rating:** 8
**Confidence:** 4

**Summary:**

The study investigates how LLMs solve multiple-choice question-answering (MCQA) tasks, focusing on intrinsic interpretability. The authors explore how the specific layers and attention heads contribute to selecting the correct answer from a set of choices. By applying methods like vocabulary projection and activation patching, the study reveals that particular middle transformer layers play a critical role in aligning the model’s predictions with the correct answer symbol. These components act selectively to bind the answer symbol (e.g., A, B, C, D) to the correct answer phrase. The paper provides a comprehensive pre-analysis of multiple models, including Olmo, Llama, and Qwen, across various prompt formats and datasets to select the models that understand the MCQA task. Additionally, it introduces a synthetic MCQA task to isolate MCQA capabilities from dataset-specific challenges. Key findings indicate that robust MCQA performance relies on both specific layers and sparse attention heads that drive answer selection, with some layers adapting to unusual symbol choices in later stages. This work offers a deeper understanding of model-specific MCQA mechanisms.

**Strengths:**

- Evaluating across various prompt formats and permutations is really convincing, it really highlights this paper
- Interesting analysis that includes both MHSA and MLP patching
- Good visualization of when the MCQA understanding forms during training

**Weaknesses:**

- The motivation for selecting the patching format is not fully substantiated. By "patching format," I refer to the reordering of the correct answer to a new position. Although this approach is intuitively clear, I believe the motivation could be strengthened by exploring different patching formats (for example, removing correct answer from sample).
- The circuits are shown to be dependent on the nodes (parts of model) for patching [1]. Patching the layers is interesting, but for full picture it would be good to examine the attention maps by patching.

[1] Miller, Joseph, Bilal Chughtai, and William Saunders. "Transformer Circuit Evaluation Metrics Are Not Robust." First Conference on Language Modeling. 2024.

**Questions:**

- Could you please specify what values are presented by Fig. 8? Is taken from logit lens applied to MSHA components by heads?
- What conclusion do you make from the fact that in Fig. 7c (MHSA) change in answers are visible only in 24 layer, while in Fig.3 it maintains further. It will be interesting to describe in detail that mechanism.
- It is relatively small issue, but do you consider using another variant of logit lens? The reason is that logit lens by itself could be potentially worse for earlier layers [2]


Small comments:
- Fig.8 is not aligned

[2] Belrose, Nora, et al. "Eliciting latent predictions from transformers with the tuned lens." (2023).

---

> ### Author Response · Authors · 2024-11-22
> **Author Response**
>
> ### Thank you for your positive comments about our work and valuable questions!
>
> > *"The motivation for selecting the patching format is not fully substantiated. By "patching format," I refer to the reordering of the correct answer to a new position. Although this approach is intuitively clear, I believe the motivation could be strengthened by exploring different patching formats (for example, removing correct answer from sample).”*
>
> As we mention in lines 231-233, preliminary conditions for obtaining meaningful interpretations from patching is that 1) the model predicts the correct answer for both patching formats, and 2) the answer changes after a patch. Removing the correct answer from the sample does not result in a valid new correct answer, making it difficult to interpret the role of various model components since the task has fundamentally changed. The two methods we use (reordering the correct answer to a new position and changing the answer choice symbols) ensure the correct answer has changed but is still valid, allowing us to isolate mechanisms for predicting a specific correct answer.
>
> > *"The circuits are shown to be dependent on the nodes (parts of model) for patching [1]. Patching the layers is interesting, but for full picture it would be good to examine the attention maps by patching.”*
>
> Good suggestion; we will include a plot for attention head patching in the updated PDF (coming soon). (We assume you mean "attention head" by "attention map" here; but please let us know if this is not what you mean).
>
> ### Your questions:
> > *"Could you please specify what values are presented by Fig. 8? Is taken from logit lens applied to MSHA components by heads?”*
>
> Yes, this is exactly what is plotted, following Yu et al (2023)’s methodology. We describe the decomposition of the MHSA component into attention heads in Appendix A.2 (lines 835-846) and briefly in lines 456-457, but we’ll add more details in the Figure 8 caption so it’s clearer.
>
> > *"What conclusion do you make from the fact that in Fig. 7c (MHSA) change in answers are visible only in 24 layer, while in Fig.3 it maintains further. It will be interesting to describe in detail that mechanism.”*
>
> This provides supporting evidence for the two-stage prediction process for the Q/Z/R/X prompt. When the correct answer symbol (A to D) and location (index 0 to index 3) change in Figure 7c, but not the vocabulary space of answer choices (A,B,C,D), layer 24 is strongly causally implicated in the model switching its prediction from A to D– this may be encoded in the residual stream at layer 24 as either positional information saying “the correct answer is at index 3” or as the actual letter, “D”, both of which flip the model’s prediction to “D”. But when only the answer symbol (A to Q) and the vocabulary space (Q,Z,R,X) change and not location (index 0 stays correct), we observe layers 27-29 playing a key role. This means that layer 24 is *neither* encoding that 1) Q represents position 0 and position 0 is correct *nor* that 2) Q represents the selected answer. Aligned with Figure 6a, Fig. 3 depicts that there is a delayed layerwise effect in assigning non-negligible probability to Q.
>
> > *"It is relatively small issue, but do you consider using another variant of logit lens? The reason is that logit lens by itself could be potentially worse for earlier layers [2]”*
>
> This is the primary reason we also employ causal tracing, as it can discover key mechanisms in early layers that logit lens cannot. We do not use the tuned lens approach [2] because it trains a linear probe instead of using the vocabulary space defined by the model’s unembedding matrix, and thus cannot answer our research question of how predictions form in the model’s vocabulary space (lines 239-241).

---

> > ### Comment · Reviewer_FfQL · 2024-11-25
> >
> > Thank you for the detailed clarifications and responses to my comments.
> >
> > Regarding the tuned lens [2], I appreciate your explanation of why it was not employed. However, I would like to clarify that the tuned lens methodology (as I believe) does indeed use the model’s unembedding matrix but applies an affine transformation beforehand. While I understand your rationale, I still believe that incorporating the tuned lens approach could add depth and potentially strengthen your paper. That said, it is not essential to your study and remains a suggestion for consideration.
> >
> > I am particularly eager to see the updated results, including the plots for attention head patching, as they promise to provide further insights into the model’s mechanisms. Thank you for your thorough and thoughtful responses, and I look forward to the revised version of your work.

---

> ### Author Response · Authors · 2024-11-30
>
> Thank you for your patience. We completed the attention patching results on each attention head for the experimental setup plotted in Figure 7(c) from layer 22 onward (i.e., where we first observe an effect from the MHSA function). The results demonstrate that the promotion of specific answer choices (the spikes in Fig. 7c) are attributed to **1-4 attention heads per layer**. To give an example, you can see the result for patching each of the 32 attention heads at layer 24 here, where heads 1 and 19 give a cumulative effect to produce Fig 7c's pattern: https://imgur.com/a/9EfbAUz
>
> We have added the graphs (depicting layers 22 to 31) as an additional figure to the Appendix and referenced this figure in our section 6 paragraph titled **"Answer symbol production is driven by a sparse portion of the network"** (unfortunately we are no longer able to update the PDF here for your viewing, but hope the above image temporarily suffices as evidence that we have done this). These results are valuable additional evidence to complement our vocabulary projection plots in Fig. 8 and demonstrate causal evidence of the unique roles of individual attention heads; thank you for the suggestion.
>
> Our current PDF version that we uploaded earlier this week includes an updated caption for Figure 8 with more details about the experiments as you requested-- please see our general response above for more information. We are happy to answer any more questions or suggestions you may have.

---

> > ### Comment · Reviewer_FfQL · 2024-12-02
> >
> > Thank you for the detailed response and additional experiments. I find the new results on attention patching more convincing and very interesting. I’ve raised my score to reflect this improvement.
> >
> > Out of curiosity (not affecting the score, I believe that the paper is already strong), how long does it take to run the head-patching experiments for a single layer? Is it even possible to provide such computation for all layers?
> >
> > Thank you again for your thorough revisions!

---

> ### Author Response · Authors · 2024-12-03
>
> Thanks for your detailed engagement with our paper & consideration of our rebuttal.
>
> For each patch, we perform 2 inference runs on GPU in parallel (i.e., on prompts $x_A$ and $x_B$) and then patch a hidden state from one run over to the other (i.e., $x_B\rightarrow x_A$) before completing the inference run on input $x_A$ to see how the score changes as a result of the patch. For layerwise analysis, this is, 32 paired inference runs over the dataset for a 32-layer model like Olmo 7B. In the naive implementation, patching each attention head adds a number-of-attention-heads multiplicative factor, so for Olmo 7B, you would now run the inference procedure over the dataset 32 heads * 32 layers number of times. This can be sped up slightly by caching the hidden states up to the layer where you are performing the intervention. There is also some nascent work on speeding patching up by using gradient-based approximation techniques (https://www.neelnanda.io/mechanistic-interpretability/attribution-patching), but we stuck to the purist approach here to avoid approximating anything.

---

### Official Review · Reviewer_E4U5 · 2024-11-07

**Soundness:** 4
**Presentation:** 4
**Contribution:** 3
**Rating:** 8
**Confidence:** 4

**Summary:**

The paper is devoted to the analysis of the internal processing of Transformer LLMs when answering Multiple Choice Question Answering task. First, it is shown that the ability to answer such question can be attributed to a few layers in the model. Second, this ability is implemented by a sparse set of attention heads. Third, this ability emerges at some point during model's training. Finally, an interesting observation is that in case of on-standard label symbols (e.g. QXYZ instead of ABCD) the model first tries to predict typical characters corresponding the answer position, which are replaced with the correct character in higher layers.

The analysis is performed on three MCQA dataset: MMLU, HellaSwag, and synthetic Colors dataset. As for models, three families are considered: LlaMa, Qwen and Olmo. For deep analysys, the models with consistent results (i.e. robust to answer order and labels perturbation) on all the three datasets are chosen

**Strengths:**

+ The paper is clearly written. All the claims of the paper are supported by experiments, and the results are sound
+ The claimed properties are consisstently observed over datasets and model's families
+ The observed results are interesting and helps to understand the internal mechanics of LLMs
+ Althougt the methods used in the paper are not novel, the quality of the presented analysis is quite high. I especially appreciate the models's robustness analysis and the difference between probits and logits

**Weaknesses:**

- The range of considered model sizes is limited (0.5B-7B)
- Although the observations presented in the paper are clear and consistent, they don't provide any real understanding how the model works. Besides, it is limited to Multiple Choic eQuestion Answering tasks, and there is no attempts to extend it to more general understanding of the inner Transformer's mechanics.

**Questions:**

1. In sec.3.4, why the threshold on 20%+ random (i.e. 45%) is chosen? Is there any reason behind this choice?

2. Evalusting 3-shot accuracy with alternative symbols (e.g. QXYZ), which symbols are used for in-context examples?

3. Do you have any ideas which of the observed phenomena can be extended to other tasks, beyond MCQA?

---

> ### Author Response · Authors · 2024-11-22
> **Author Response**
>
> ### Thanks for all of your positive feedback!
> > *"The range of considered model sizes is limited (0.5B-7B)”*
>
> The only other sizes of Llama 3.1 available are 70B and 405B, and Olmo does not have any model sizes apart from those we tested (1B and 7B). We included Qwen 2.5 0.5B and 1.5B sizes in the paper to provide a size contrast to the 7B and 8B models; we feel these 3 model families and 5 sizes are fairly representative.
>
> While we unfortunately did not have the resources to scale to super large (70B+) sizes, we note this is the case in virtually all current academic mechanistic interpretability research. However there are some promising up-and-coming initiatives (such as https://arxiv.org/abs/2407.14561) to allow for analysis at larger scales soon, so we hope we can scale in future work.
>
> ### Your questions:
> > *"why the threshold on 20%+ random (i.e. 45%) is chosen?”*
>
> Good question. 20% represented a reasonable trade-off to us between selecting high-performing models and having a sufficiently representative sample of models, but the exact threshold doesn't matter. We’ll reword this in the updated PDF (coming soon) to be that we selected the best-performing model from each model family for further analysis, and they all do all 3 tasks sufficiently above random (in this case, >20% above random).
> > *"Evalusting 3-shot accuracy with alternative symbols (e.g. QXYZ), which symbols are used for in-context examples?”*
>
> We use the same alternative symbols for the in-context examples. See footnote 4 on line 214.
> > *"Do you have any ideas which of the observed phenomena can be extended to other tasks, beyond MCQA?”*
>
> This is an open question! We don’t want to extrapolate too much, but there is some initial evidence that models share circuit components across tasks (such as https://arxiv.org/abs/2310.08744). This is an important direction for future work.

---

### Author Response · Authors · 2024-11-27
**General Response + Updated PDF**

Thank you for engaging with us during the rebuttal period, and for your valuable comments! We have made the following changes to the PDF (marked in red for your easy viewing):

Writing:
- Changed language about 20%-above-random model selection threshold in the final paragraph of Section 3.4 (Reviewer **E4U5**)
- Fixed typos and format of some citations (Reviewer **xW8X**)
- Updated the last sentence of the abstract and the last paragraph of Section 7 to soften the claim about poor performing models (Reviewer **xW8X**)
- Updated the caption of Fig. 1 for clarity (Reviewer **xW8X**)
- Added model accuracy #s to Fig. 4 (Reviewer **xW8X**)
- Provided additional details about the metrics in Fig. 8 at the end of Section 6 (Reviewer **xW8X**), updated caption for clarity and made subfigures aligned (Reviewer **FfQL**)
- Added some discussion on the strengths and weaknesses of causal tracing vs. vocab projection in Appendix A.3 (Reviewer **WzgP**)

Experiments/results:
- To further support our claim that one means by which models produce OOD answer choice symbols is by first operating in the space of familiar answer tokens (A/B/C/D), we have done the following (suggested by Reviewer **xW8X**):
1. We added experiments on the Q/Z/R/X prompt for 2 additional models to the Appendix: Figs. 20 and 21 for vocabulary projection, Fig. 15a + 15b for causal tracing. We observe that Qwen 2.5 1.5B has a similar pattern as Olmo 7B Instruct but Llama 3.1 does not, indicating that the effect is not unique to Olmo, but also not general.
2. We added experiments on another random set of letters (“OEBP”) for Olmo 7B Instruct in Figure 21 and Figure 15c. Comparing to the first row of Fig. 6, the patterns are largely similar, indicating that the finding is **not contingent on a specific random-letter prompt**.
3. We updated the text in various places (marked in red, particularly the last 2 paragraphs of Section 5) to reflect these results.

- We updated Figs. 9, 21, 22 (now Figs. 9, 26, 27) to be 3-shot instead of 0-shot, for consistency with the rest of the paper (Reviewer **WzgP**). The observed trends are largely similar to the previous 0-shot version, which may be due to the fact that adding in-context examples barely changes performance for this model (+4% MMLU accuracy, -2% HellaSwag, Colors stays the same).

- Due to some compute constraints, we’re still working on adding the plots for attention head patching requested by reviewer **FfQL** and will update again later.

---

> ### Comment · Reviewer_xW8X · 2024-12-02
>
> New results for Qwen 2.5 and LLaMA 3.1 are very interesting. I wonder, why LLaMA behaves differently, than OLMO and Qwen in the experiments with Q/Z/R/X symbols for answer choices. Maybe Qwen and OLMO were over-tuned on some MCQA tasks with A/B/C/D letters, and this is why they "want" to assign some scores to these "familiar" letters before going to Q/Z/R/X, and LLaMA is less "familiar" with such tasks and thus less inclined to A/B/C/D format? I don't know for sure, it's just a hypothesis.
>
> Anyway, I increased my score from 5 to 6 due to new results that make paper more valuable.

---

> > ### Author Response · Authors · 2024-12-03
> >
> > Thanks for your engagement with our paper & rebuttal. It's an interesting hypothesis you put forward, and one we hope to understand better in future work.

---

### Meta-Review · Area_Chair_PvfQ · 2024-12-20

**Metareview:**

This paper investigates how LLMs solve Multiple Choice Question Answering (MCQA) tasks through mechanistic interpretability analysis. The key findings show that specific middle transformer layers and sparse attention heads play critical roles in selecting correct answers, with some models using a two-stage process when handling non-standard answer symbols (e.g., Q/Z/R/X instead of A/B/C/D). The paper's main strengths lie in its comprehensive evaluation across multiple models (Olmo, Llama, Qwen) and datasets (MMLU, HellaSwag, Colors), robust experimental design including permutation tests, and clear visualization of results. While the work is limited to relatively small models (0.5B-7B parameters) and focused specifically on MCQA rather than broader model mechanics, the thorough analysis and consistent findings across different models make this a valuable contribution to understanding LLM behavior. The paper merits acceptance due to its sound methodology, clear presentation, and important insights into how transformer models process structured multiple choice tasks.

**Additional Comments On Reviewer Discussion:**

The discussion period led to several improvements and clarifications. Reviewer FfQL requested attention head patching experiments to complement the layer-wise analysis, which the authors addressed by adding new figures showing individual attention head contributions. Reviewer xW8X questioned the generalizability of the two-stage prediction process finding, leading the authors to add experiments with additional models (Qwen 2.5, Llama 3.1) and alternative answer symbols (OEBP), strengthening their claims. Reviewer WzgP raised concerns about using 0-shot versus 3-shot examples in certain experiments, which the authors addressed by updating multiple figures for consistency. The reviewers generally responded positively to these changes, with reviewer FfQL and xW8X explicitly increasing their scores after seeing the additional results. Several reviewers maintained high confidence in their assessments throughout the discussion. The addition of attention head patching and generalization experiments reinforces this paper.

---

### Decision · Program_Chairs · 2025-01-22

Accept (Spotlight)